# Accelerating discovery of bioactive ligands with pharmacophore-informed generative models

Weixin Xie[1,6], Jianhang Zhang[2,6], Qin Xie[2], Chaojun Gong[2], Yuhao Ren[3], Jin Xie[1], Qi Sun ✆[3,4,5], Youjun Xu ✆[2] ✉, Luhua Lai ✆[1,3,4,5] ✉ & Jianfeng Pei ✆[1,5] ✉

Deep generative models have advanced drug discovery but often generate compounds with limited structural novelty, providing constrained inspiration for medicinal chemists. To address this, we develop TransPharmer, a generative model that integrates ligand-based interpretable pharmacophore fingerprints with a generative pre-training transformer (GPT)-based framework for de novo molecule generation. TransPharmer excels in unconditioned distribution learning, de novo generation, and scaffold elaboration under pharmacophoric constraints. Its unique exploration mode could enhance scaffold hopping, producing structurally distinct but pharmaceutically related compounds. Its efficacy is validated through two case studies involving the dopamine receptor D2 (DRD2) and polo-like kinase 1 (PLK1). Notably, three out of four synthesized PLK1-targeting compounds show submicromolar activities, with the most potent, IIP0943, exhibiting a potency of 5.1 nM. Featuring a new 4-(benzo[*b*]thiophen-7-yloxy)pyrimidine scaffold, IIP0943 also has high PLK1 selectivity and submicromolar inhibitory activity in HCT116 cell proliferation. TransPharmer offers a promising tool for discovering structurally novel and bioactive ligands.

Identifying compounds with bioactivity against desired targets has been one of the important objectives for rational drug discovery. Deep learning-based generative models have emerged as currently predominant methodologies, demonstrating their efficacy in advancing towards this objective[1–13]. One well-known instance is that scientists at Insilico Medicine successfully employed their generative model, GENTRL, to uncover nanomolar inhibitors for the DDR1 kinase within a short timeline[1]. Beyond GENTRL, researchers exhibit a fervent interest in exploring the potential of molecular generative models through investigations of diverse combinations of model components, including architectures[14–19], molecular representations[20–22], and optimization algorithms[23–26].

Effective as generative models are, their efficiency raises new concerns: how does the creativity of generative models compare to that of humans? Can the designs generated by these models inspire human experts? In 2018, Bush et al. conducted an interesting experiment—a Turing test involving three molecular generators[27], including RG2Smi, a deep learning-based generative model[28]. They found that it was hard for RG2Smi to propose molecular designs that align with those of human medicinal chemists or gain acceptance from them. Moreover, the novelty of the bioactive compounds generated automatically has constantly been under debate[29–31]. Moret et al. fine-tuned their chemical language models (CLMs) using 46 highly active PI3Kγ inhibitors before employing them to generate new inhibitors against

[1]Center for Quantitative Biology, Academy for Advanced Interdisciplinary Studies, Peking University, Beijing, China. [2]Infinite Intelligence Pharma, Beijing, China. [3]BNLMS, Peking-Tsinghua Center for Life Sciences at the College of Chemistry and Molecular Engineering, Peking University, Beijing, China. [4]Peking University Chengdu Academy for Advanced Interdisciplinary Biotechnologies, Chengdu, China. [5]Research Unit of Drug Design Method, Chinese Academy of Medical Sciences, Beijing, China. [6]These authors contributed equally: Weixin Xie, Jianhang Zhang. ✉e-mail: xuyj@iipharma.cn; lhlai@pku.edu.cn; jfpei@pku.edu.cn

PI3Kγ kinase. The chemical structures of the most potent ligands designed or inspired by CLMs, namely compounds **18** and **22**, exhibit a high degree of similarity to known PI3Kγ inhibitors[2]. Other studies that applied transfer learning to bias molecular generators toward specific protein targets often encounter varying degrees of novelty issues with the bioactive compounds generated[3-6]. These results underscore the urgent need for a deep understanding of the "correct recipes" for generative models to produce compounds that are bioactive while novel enough, in order to serve as useful copilots for human medicinal chemists.

Pharmacophore-informed generative models present alternative approaches to promote this understanding. The pharmacophore model, rooted in pharmaceutical features, offers a coarse-grained solution for molecular representation, facilitating scaffold hopping among chemically diverse ligands[32,33]. Furthermore, pharmacophore serves as a bridge linking molecular structure and bioactivity. Given these advantages, there has been a recent surge in interest regarding the utilization of pan-pharmacophore features for molecular generation[34-36]. For instance, Imrie et al. introduced DEVELOP, a pharmacophore-aware generative model employing 3D grids to represent target pharmacophores, for linker design and scaffold elaboration[34]. Their results demonstrated that generative models can leverage pharmacophoric information to produce molecules with distinct structures that maintain crucial non-bond interactions with receptors. Similarly, LigDream encodes and decodes 3D voxels representing five common types of pharmacophore features for de novo molecular design[35]. Other pan-pharmacophore features have been incorporated into generative models, including condition vectors indicating the shortest bond distances, as well as the presence, absence, or exact quantities of specific pharmacophoric features[37,38]. Recently, Zhu et al. introduced PGMG, which employs a fully connected graph containing selected pharmacophore features of a reference compound[39]. PGMG was able to generate drug-like molecules with superior docking scores compared to known bioactive ligands and showcased its capability of scaffold hopping from an initial EGFR inhibitor. However, it is noteworthy that most novel molecules generated by pharmacophore-based generative models have not yet undergone wet lab experimental testing to validate this methodology.

In this study, we present TransPharmer as an innovative pharmacophore-aware generative model, which employs ligand-based pharmacophore kernels to achieve structural abstraction while preserving fine-grained topological information. The ligand-based pharmacophore kernels are similar to those used in the previous studies for ligand-based virtual screening[40,41]. Our pharmacophore kernels are encoded into multi-scale and interpretable fingerprints, serving as prompts for TransPharmer. The architecture of TransPharmer is reminiscent of a generative pre-training transformer (GPT)[42], as illustrated in Fig. 1, establishing a connection between pharmacophores and molecular structures represented by the simplified molecular-input line-entry system (SMILES)[43]. We posit that equipping GPT with pharmacophore knowledge enables the model to focus on the pharmaceutical aspects of the chemical structures and generate drug-like molecules. During our evaluation, TransPharmer demonstrated superior performance compared to other baseline models in tasks involving de novo generation and scaffold elaboration under pharmacophoric constraints. We also highlight TransPharmer's distinct mode in probing the local chemical landscape surrounding a reference compound, rendering it highly suitable for scaffold-hopping tasks in drug discovery. We further validate the capability of TransPharmer to produce innovative and bioactive ligands through two case studies involving DRD2 and PLK1. Notably, we experimentally tested four generated compounds targeting PLK1, which feature a new series of scaffolds. Among these, three out of four compounds exhibit inhibitory activity below 1 μM, with the most potent one, IIP0943, demonstrating a potency of 5.1 nM (4.8 nM for the reference PLK1 inhibitor). Furthermore, IIP0943 exhibits high selectivity for PLK1 compared to other Plks and submicromolar activity in cell proliferation against the HCT116 cell line. TransPharmer thus represents a pharmacophore-based generative model successfully executing scaffold hopping to produce unique compounds with potent bioactivity. The 4-(benzo[*b*]thiophen-7-yloxy)pyrimidine scaffold of IIP0943 may offer new insights for obtaining improved PLK1 inhibitors.

## Results

In this work, we developed a pharmacophore-based generative model named TransPharmer, which leverages the topological pharmacophore fingerprints of given ligands to guide molecule generation. The workflow and model setups are shown in Fig. 1 and detailed in subsections "Pharmacophore features and fingerprint extraction" and "Model architecture".

We observed that the pharmacophore fingerprints employed in our study have the potential to establish connections between structurally distinct ligands that exhibit activity towards the same target. Additionally, these fingerprints demonstrate a notable relationship with bioactivity, allowing for the distinction between active and inactive ligands (Supplementary Notes subsection "Pharmacophore fingerprints as fuzzy and interpretable representations"). The unconditional version of TransPharmer demonstrates accurate modeling of chemical space, achieving the top rank among established methods in overall performance in the GuacaMol benchmark[44], and achieving top 2 ranks in six out of fifteen metrics benchmarked in MOSES[45] (Supplementary Notes subsection "Benchmarking the unconditional TransPharmer and other evaluations").

The results in this section are organized as follows: first, we evaluate the performance of TransPharmer on two tasks involving pharmacophore-constrained molecule generation. Secondly, we compare the unique mode of chemical space exploration of TransPharmer with a previous method based on structure mutations. Thirdly, we demonstrate the capability of TransPharmer to generate active ligands through a retrospective case study of recalling known DRD2 actives distinct from the ones seen during training. Lastly, we highlight the potential of TransPharmer in a prospective case study for discovering potent and highly selective PLK1 inhibitors with scaffolds different from previous ones.

### Pharmacophore-constrained molecule generation

One of the central objectives for pharmacophore-conditioned generative models is to generate molecules conforming to the desired pharmacophores, which entails two aspects. Firstly, basic attributes of the pharmacophores of the generated molecules should match those of the target, such as the number of individual pharmacophoric features. Generating molecules with the requisite number of pharmacophoric features has been an essential objective[37,38,46]. Here, we computed the averaged difference in the number of individual pharmacophoric features of generated molecules with respect to the target pharmacophores (referred to as $D_{\mathrm{count}}$, see definition in Section "Evaluation metrics"). Secondly, the targeted pharmacophore and the generated molecule's pharmacophore should have a high degree of overall similarity. Similar to measuring molecular similarity using fingerprints such as Morgan fingerprints[47], pharmacophoric similarity can be calculated by computing the Tanimoto coefficient of two pharmacophoric fingerprints. Here, we adopt ErG fingerprints implemented in RDKit[48] to measure pharmacophoric similarity (referred to as $S_{\mathrm{pharma}}$, see definition in Section "Evaluation metrics") to avoid any artificial positive results of our models. ErG fingerprints are another pharmacophoric fingerprint introduced by researchers in Lilly and have demonstrated potential applications for scaffold hopping[32]. ErG fingerprints show a discernible correlation with the pharmacophoric fingerprints utilized in TransPharmer (Supplementary Fig. 1).

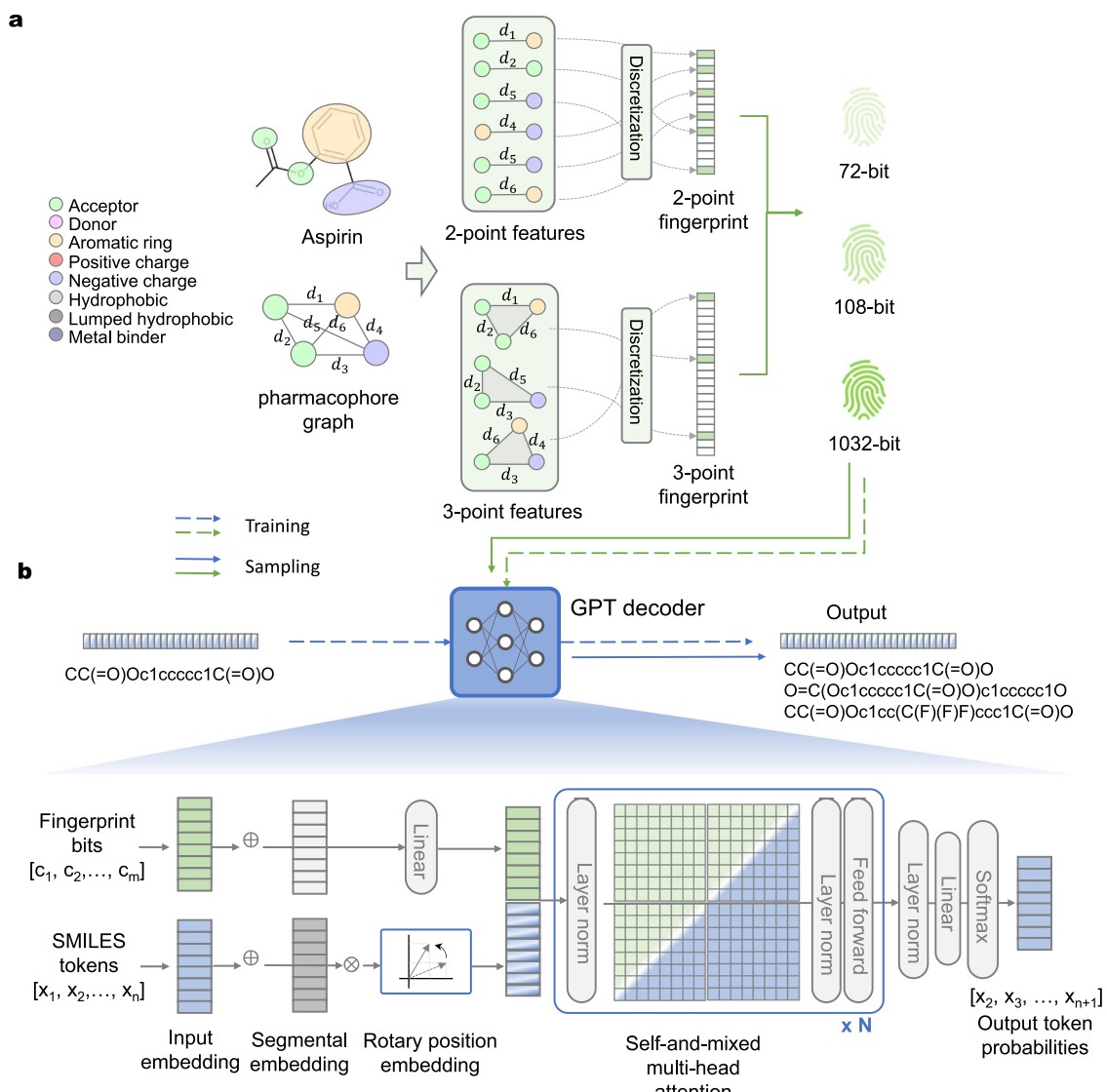

**Fig. 1 | The schematic diagram of TransPharmer architecture. a** The process of pharmacophore fingerprint extraction. As an example, the chemical structure of Aspirin is converted into a pharmacophoric topology graph with the shortest topological distance between each feature pair computed. All the two-point and three-point pharmacophoric subgraphs are enumerated, and the topological distances are discretized with specific distance bins. 72-bit and 108-bit pharmacophore fingerprints are constructed from the two-point pharmacophores with different discretization schemes, while 1032-bit pharmacophore fingerprints are the concatenation of fingerprints of two-point and three-point pharmacophores. **b** The architecture of TransPharmer as a pharmacophore fingerprints-driven GPT decoder.

We compare our models with LigDream[35], PGMG[39], and DEVELOP[34] as baselines in both tasks of de novo generation and scaffold elaboration. As pharmacophoric feature counts have been utilized as explicit controls over the generated molecules[37,38,46], we establish another baseline by training a "TransPharmer-count" model that only accepts the requirement of desired amounts of individual features. Furthermore, to investigate the effect of the length of pharmacophoric fingerprints used in our model, three variants of TransPharmer were examined: "TransPharmer-72bit", "TransPharmer-108bit" and "TransPharmer-1032bit". These variants are conditioned on 72-bit, 108-bit, and 1032-bit pharmacophoric fingerprints, respectively.

For the de novo generation task (Table 1), TransPharmer models outperform the baseline models by generating molecules with higher pharmacophoric similarity. It is noteworthy that the TransPharmer-count model achieves the lowest deviation in feature counts, while the TransPharmer-1032bit model ranks as the second lowest in this regard. It is not directly comparable between PGMG and other methods since PGMG is primarily designed to align with a specific subset of pharmacophore features (specifically, 3–7 features), whereas models such as TransPharmer aim to generate molecules that satisfy the entire set of pharmacophore features of a reference compound. Consequently, We re-evaluated TransPharmer based on the match score utilized in PGMG and discovered that the match scores achieved by TransPharmer are close to those of PGMG, with the smallest difference being less than 10% of PGMG's score (Supplementary Table 1). Meanwhile, it is worth noting that PGMG is sensitive to the maximum number of input pharmacophore features specified by users, which leads to a notable deviation in molecular sizes compared to the desired targets, particularly when reference compounds possess flexible conformations (Supplementary Table 2). Further discussion can be found in the Supplementary Notes.

In the scaffold elaboration task, four TransPharmer models generated molecules with substantially higher pharmacophoric similarity than those of DEVELOP. It seems that DEVELOP exhibits limitations in adhering to the provided pharmacophore conditions, often resulting in the generation of molecules unrelated to and much larger than the reference compound (Supplementary Tables 3 and 4). Among the four TransPharmer models evaluated, the TransPharmer-1032bit model

**Table 1 | Results of the pharmacophore-constrained de novo generation task and scaffold elaboration task**

| Method | De Novo generation | | Scaffold elaboration | |
|---|---|---|---|---|
| | $D_{count}$ | $S_{pharma}$ | $D_{count}$ | $S_{pharma}$ |
| LigDream | 4.1 ± 2.6 | 0.47 ± 0.14 | n.a. | n.a. |
| PGMG | 9.4 ± 3.7 | 0.35 ± 0.13 | n.a. | n.a. |
| DEVELOP | n.a. | n.a. | 13.0 ± 7.0 | 0.231 ± 0.160 |
| TransPharmer-count | **0.3 ± 0.4** | 0.48 ± 0.13 | **0.2 ± 0.3** | 0.706 ± 0.218 |
| TransPharmer-72bit | 4.6 ± 2.6 | 0.50 ± 0.14 | 3.0 ± 2.2 | 0.702 ± 0.176 |
| TransPharmer-108bit | 3.6 ± 2.6 | 0.58 ± 0.15 | 2.3 ± 1.9 | 0.751 ± 0.167 |
| TransPharmer-1032bit | 3.3 ± 2.0 | **0.60 ± 0.14** | 2.1 ± 1.7 | **0.754 ± 0.166** |

The evaluation of LigDream and PGMG only focused on the de novo generation task, while DEVELOP was assessed solely on scaffold elaboration, which aligns with their original development purposes. Therefore, n.a. (not applicable) is assigned to the performance of LigDream and PGMG in scaffold elaboration and DEVELOP in de novo generation tasks, respectively. Feature count deviation ($D_{count}$) and pharmacophoric similarity ($S_{pharma}$) scores were computed with respect to the conditioning compounds. TransPharmer-count refers to the TransPharmer model conditioned on the required number of eight types of pharmacophoric features. TransPharmer-72bit, TransPharmer-108bit, and TransPharmer-1032bit refer to the TransPharmer models conditioned on pharmacophoric fingerprints of lengths 72, 108, and 1032, respectively. The numbers in bold indicate the best values.

achieves the highest similarity score. The TransPharmer-count model is slightly better than the TransPharmer-72bit model in the mean pharmacophoric similarity, but the variance is larger. The deviation of feature counts is similar to those in the de novo generation task.

These findings suggest the benefits of employing pharmacophoric fingerprints that explicitly encode the topology of pharmacophores. In comparison to the 3D voxels of pharmacophoric points encoded by convolutional layers in LigDream or DEVELOP, pharmacophoric fingerprints offer more distinct instructions for molecule generation and may avoid ambiguous guidance resulting from insufficient training of the convolutional neural networks. When compared to simplified condition vectors like feature counts, pharmacophoric fingerprints encompass comprehensive information regarding the topology of pharmacophores, thereby providing superior guidance. In contrast to the use of pharmacophore graphs of selected features in PGMG, TransPharmer exhibits superior control over global molecular properties, such as molecular weight and the number of heavy atoms, resulting in improved sampling efficiency. An ablation study further showed that the incorporation of topological distance information and feature combinations into the pharmacophore fingerprint substantially contributes to TransPharmer's overall performance (Supplementary Notes "Ablation study" and Supplementary Table 15). For the de novo generation task, removing the topological distance information decreased the pharmacophore similarity score from 0.50 to 0.38, and removing both the topological distance information and feature combinations further decreased it to 0.31. For the scaffold elaboration task, removing both the topological distance information and feature combinations decreased the pharmacophore similarity score from 0.70 to 0.55.

Our analysis also reveals that TransPharmer models with longer pharmacophoric fingerprints consistently generate molecules with higher similarity to the target pharmacophore (Table 1), which conforms to our motivation to obtain fine-grained representations of pharmacophore. Moreover, these models generated molecules that were more similar to the conditioning compound in terms of topological structure and had a relatively higher repetition rate (Supplementary Table 5). Depending on specific needs, the flexibility of TransPharmer allows users to choose which model is most suitable for their intended applications. Overall, the excellent performance and

flexibility of TransPharmer make it a viable option for a wide range of scenarios such as novel hits discovery or lead optimization.

## Exploring local chemical space

Efficiently exploring the vast chemical space remains a challenging task in drug discovery. One common approach is to start with a few compounds and search their neighborhood but the exploring direction can be quite arbitrary. Molecular similarity-constrained exploration/optimization is one of the widely adopted ways to identify compounds with the desired similarity level to the starting compound[49–52]. In the previous section, we demonstrated that TransPharmer can efficiently explore the local chemical space in a pharmacophore-constrained fashion. Here, we compare the exploring mode of TransPharmer with those of molecular similarity-constrained methods, using a specific starting compound as a showcase, and illustrate the significance of this exploring mode in drug discovery.

We used Onvansertib, a known inhibitor of PLK1[53], as the starting compound to provide a target pharmacophore. STONED[51], which can perform molecular similarity-constrained exploration by altering the given compound structure, was used for comparison. STONED can rapidly traverse the target neighborhood in the chemical space by mutating the characters of the SELFIES string of the starting compound. Apart from the default setting, STONED can be tuned to produce highly similar structures to the starting compound by restricting the mutation area to the terminal 10% interval of the SELFIES string.[51] STONED in the default and tuned settings are referred to as "STONED" and "STONED-focused", respectively, and the details of each setting can be found in Section "The settings of compared methods". We evaluated five models, including STONED in two settings and three TransPharmer models (72-bit, 108-bit, and 1032-bit), by sampling 10,000 non-duplicate chemical structures and obtaining their pharmacophoric similarity and molecular similarity distributions with respect to the starting compound. The molecular similarity is given by the Tanimoto coefficient of Morgan fingerprints with a radius of 2 implemented by RDKit[48].

Figure 2 shows that the molecular and pharmacophoric similarity scores of the generated molecules from STONED tend to approach the same ends of the scoring range, while those of TransPharmer-72-bit are distributed near the opposite sides (Fig. 2b, e). In other words, molecules generated by TransPharmer-72-bit can be topologically dissimilar but pharmacophorically similar to the starting compound, whereas molecules from STONED are either similar in both molecular structure and pharmacophore to the starting compound, or dissimilar in both aspects (see some examples in Fig. 2g). TransPharmer can also produce structurally and pharmacophorically similar structures by using more fine-grained fingerprints (Fig. 2c, f).

The plot of the local chemical space spanned by the two similarity axes with the averaged scores of each model marked in the corresponding places in Fig. 2 illustrates that TransPharmer and STONED explore the chemical space in different directions and regions (Fig. 2d). Molecular similarity constrained methods like STONED traverse along the diagonal, while pharmacophore constrained methods like TransPharmer traverse along a line that is close to horizontal. In addition to providing new directions to explore, TransPharmer models have a unique potential to discover structurally distinct molecules while maintaining high pharmacophoric similarity (at the bottom right corner in Fig. 2d), which is essential for molecular optimization in practice, such as scaffold hopping.

## Case study of DRD2

DRD2 is a well-studied target for which many active compounds have been reported. Although ligands with known bioactivities exist, the pursuit of novel ligands with improved characteristics, such as better binding affinity or ADME/T properties, remains ongoing. Therefore, it is essential for generative models to be able to discover active

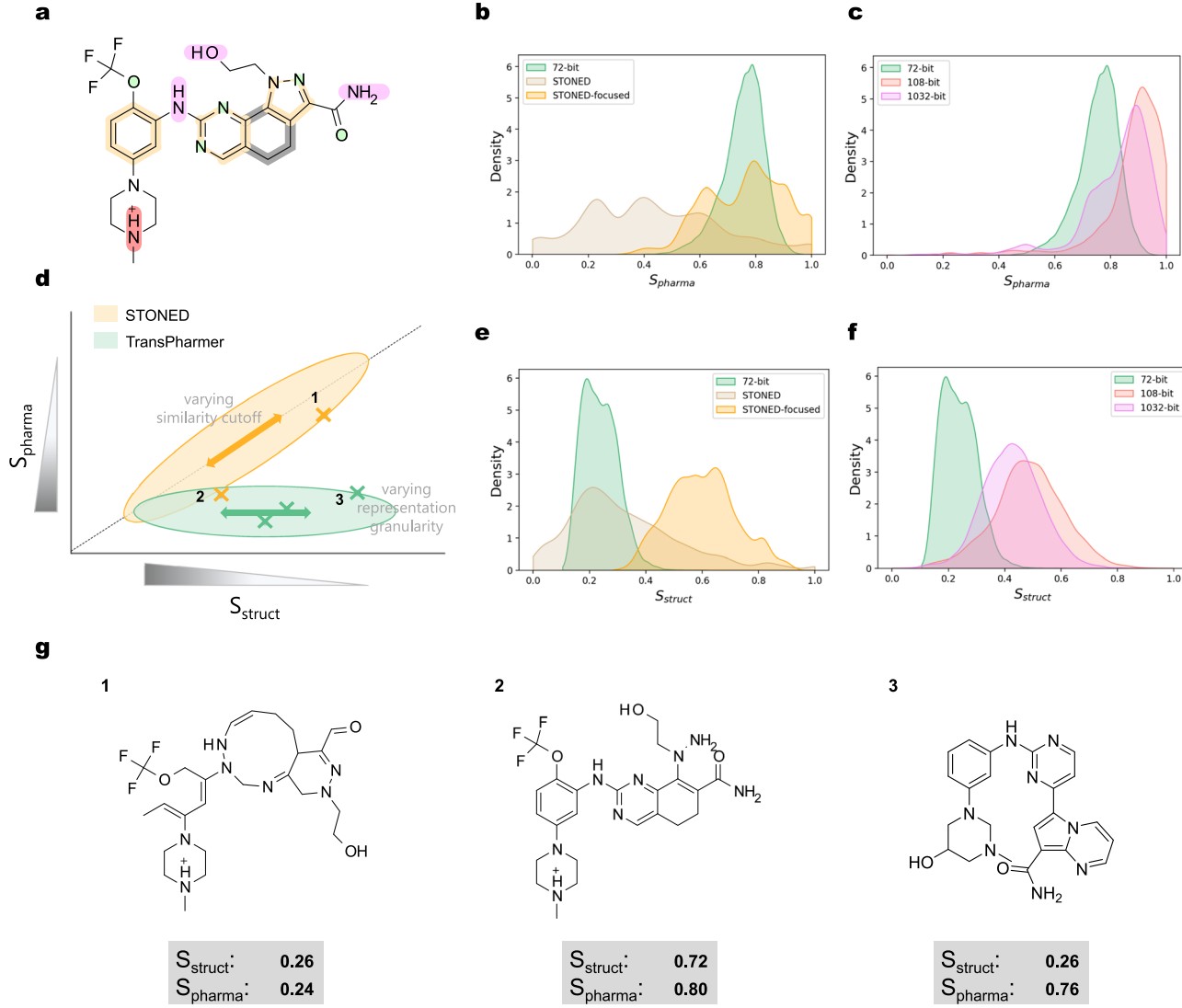

**Fig. 2 | Chemical space exploration around Onvansertib. a** The 2D chemical structure diagram with pharmacophoric features of Onvansertib. **b, e** Distributions comparison of pharmacophoric similarity $S_{pharma}$ and molecular similarity $S_{struct}$ for 72-bit TransPharmer, STONED, and STONED-focused. **c, f** Distributions comparison of pharmacophoric similarity $S_{pharma}$ and molecular similarity $S_{struct}$ for 72, 108, and 1032-bit TransPharmer models. **d** A schematic diagram of different exploration modes of pharamcophore- and molecular topology-constrained approaches in the local chemical space spanned by $S_{struct}$ and $S_{pharma}$. Cross markers label the relative positions of the mean of score distributions for the 5 models in (**b**, **c**, **e**, and **f**). Representative samples from areas 1, 2, and 3 are shown in (**g**). Source data are provided as a Source Data file.

ligands with novel structures, unrestricted by previously observed ligands.

A retrospective experiment was conducted to assess TransPharmer's ability to discover distinct and active ligands. Known DRD2 active ligands were divided into two subsets using scaffold clustering (see Section "Settings for DRD2 recall experiment"), with an average molecular similarity across these subsets of around 0.2. One subset is visible to TransPharmer during training, while the other subset is excluded from the training set. Upon completion of the training, active ligands in the training set were encoded into 72-bit pharmacophoric fingerprints and used by TransPharmer as "active conditions" for molecule generation. The retrieval of the reserved active ligands was examined. This experimental setup mimics a common but challenging scenario in drug discovery to uncover bioactive ligands possessing novel scaffold series given the known active ligands. For comparison, another set of unrelated molecules to DRD2 from the training set were used as "baseline conditions" by TransPharmer (Fig. 3b). The comparison between using DRD2 actives as conditions (active conditions)

and using baseline conditions aims to demonstrate the difficulty of this task and the consistency of TransPharmer.

The performance of TransPharmer to retrieve active ligands in the reserved subset was evaluated in two aspects. Firstly, the recall rate was calculated for all generated molecules, demonstrating the maximum potential of TransPharmer to discover active ligands under ideal conditions. However, considering the limited budgets for experimental testing, in reality, the precision of generative models is also important. In this context, we assessed the (apparent) precision of TransPharmer by enumerating active ligands found within a smaller set of repeatedly generated molecules, specifically 4000 molecules in this experiment. These molecules were generated with a higher sampling probability, indicating a greater confidence for TransPharmer to produce them during the initial sampling phase. The precision is apparent because we only search for known active ligands within the generated set, and the remaining portion likely contains potentially active ligands. Note that 4000 was chosen to be comparable to the number of active ligands unseen by TransPharmer.

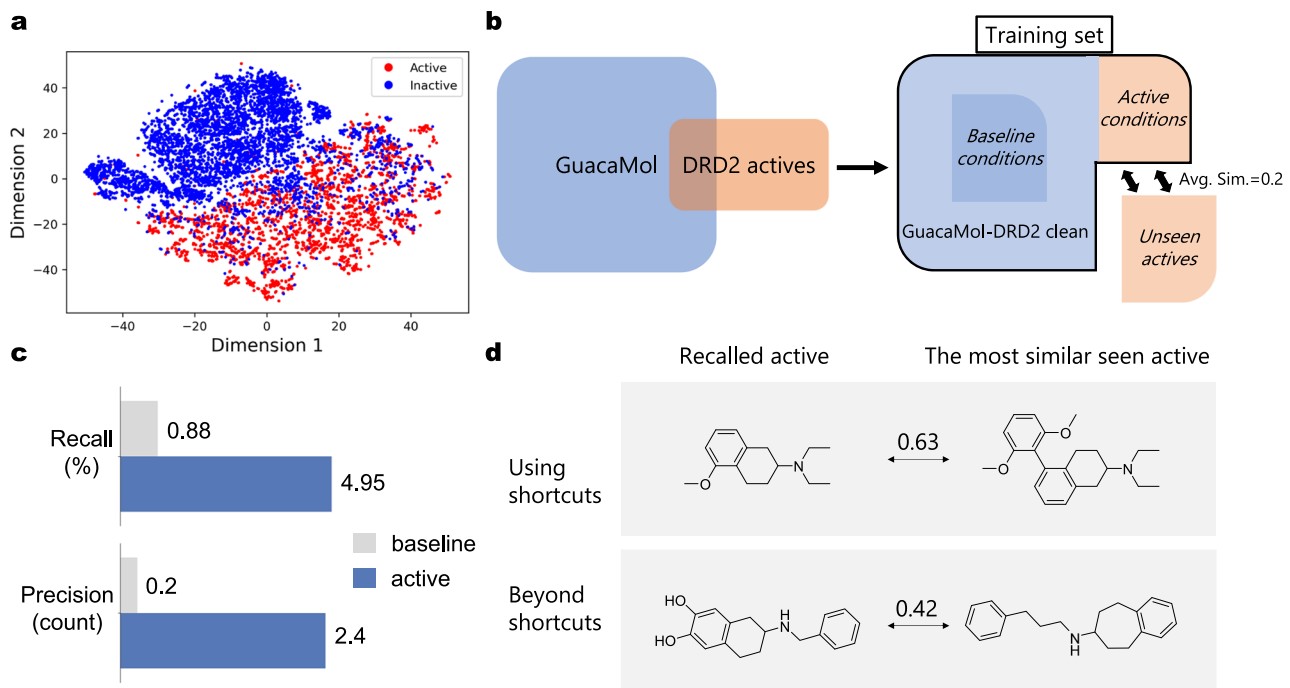

**Fig. 3 | The retrospective experiment for DRD2. a** A t-SNE plot of 108-bit pharmacophore fingerprints of 7939 active and 7939 inactive ligands of DRD2. **b** Data preparation in this experiment. "GuacaMol-DRD2 clean" refers to the subset of the original GuacaMol dataset with every known active ligand of DRD2 excluded. Baseline conditions consist of 3717 compounds randomly sampled from "GuacaMol-DRD2 clean". DRD2 actives were divided into two subsets by scaffold clustering (see Section "Settings for DRD2 recall experiment"). One subset consists of 3717 ligands of DRD2, which are visible to TransPharmer during training and used as active conditions during generation, while the other subset contains 4222 actives to be rediscovered. **c** The recall rate and precision count of generated molecules using active and baseline conditions. **d** Some recalled active ligands and their most similar counterparts TransPharmer has seen during training. An example of recalling active ligands using shortcuts is shown on the top panel while an example of recalling actives beyond any obvious shortcuts is shown at the bottom panel. The molecular similarity between the recalled and seen ligand is shown. Source data are provided as a Source Data file.

Our model rediscovered 4.95% of the active ligands in the unseen subset with sufficient sampling when conditioned on seen active ligands, compared to 0.88% when using baseline conditions (Fig. 3c). If generated molecules highly similar (Tanimoto similarity over 0.8) to any of the active ligands in the unseen subset are considered successful recalls as well, a recall rate of up to 12.1% is observed, consistently higher than that of using baseline conditions (3.2%). (Supplementary Table 6). As for the precision number, up to 15 active ligands in the unseen subset were recalled with a molecular similarity requirement of ≥0.8, which is 7-fold higher than that of using baseline conditions (Supplementary Table 6). Upon inspecting some recalled active ligands and their most similar counterparts in known active ligands, we observed that in some cases TransPharmer appeared to take shortcuts, such as borrowing subgraphs from seen molecules or making modifications based on them. However, TransPharmer was also able to rediscover ligands that are structurally distinct from any active ligands it had seen (Fig. 3d).

As previously stated, our search is limited to known active ligands within the generated set and the remaining portion likely contains potentially active ligands. This approach provides a conservative estimate of the proportion of generated molecules that exhibit activity towards DRD2. In order to obtain a more precise estimation, we conducted a virtual screening experiment following the DeepDrugCoder[54]. A DRD2 predictive model was established to predict the probability of a generated molecule exhibiting bioactivity towards DRD2 (more details in Section "DRD2 QSAR model"). We then randomly sampled 100 known active compounds from the reserved test set of the QSAR model. These compounds were used as conditions by TransPharmer to sample 256 times per active compound. The fraction of 25,600 generated SMILES strings that are valid, unique, and predicted to be active (with a predicted probability ≥ 0.5) was then computed to compare with the results of DeepDrugCoder.

We found that 27% of the generated molecules were predicted as actives, while DeepDrugCoder's physchem-based (PCB) model reported a fraction of 54%, and the fingerprint-based (FPB) model reported 19%. Since the PCB model was trained with the additional information about prior predicted bioactivity from the same QSAR model, the high ratio of molecules predicted to be active is not surprising. On the other hand, our model outperforms the FPB model in terms of ratio by over 40%, emphasizing the importance of using pharmacophoric information to identify active compounds. For molecules predicted to be active but not previously identified as DRD2 actives, we assessed structural similarities to their nearest DRD2 active neighbor. The similarity score distribution peaks around 0.4, with 43% of the molecules having a similarity score below 0.4, a commonly used threshold for classifying dissimilar compounds (Supplementary Fig. 2). This suggests a high degree of structural novelty among the generated molecules compared to known DRD2 actives. Overall, these findings highlight TransPharmer's capability to both rediscover known active ligands and to generate structurally distinct compounds with potential bioactivity.

## Case study of PLK1

PLK1 plays a key role in mitosis progression and has been implicated in various cellar pathways[55–59]. Targeting PLK1 has emerged as a promising therapeutic strategy for cancer treatment, as the over-expression of PLK1 has been associated with tumor development and progression[60,61]. In this section, we exemplify the application of TransPharmer in the generation of distinct and active PLK1 inhibitors using the topological pharmacophore fingerprint derived from Onvansertib, a potent and selective inhibitor to PLK1 currently

undergoing clinical trials (e.g., ClinicalTrials.gov identifier NCT03829410).

One million samples were generated by TransPharmer conditioned on the 72-bit pharmacophore fingerprint of Onvansertib under a low-temperature hyperparameter of 0.7. Subsequent to the removal of invalid SMILES strings and duplicated molecules, a total of 178,103 unique molecules were obtained. To gain insights into the chemical space covered by the training set, the generated molecules (both conditionally and unconditionally), and the known PLK1 active ligands, a t-distributed stochastic neighbor embedding (t-SNE) plot was generated. As shown in Fig. 4a, TransPharmer shifted from the broader chemical space of the training set to the localized chemical space surrounding Onvansertib, which appears as an "outlier", drifting apart from other PLK1 active ligands and the majority of training molecules. The similarity distributions between the generated molecules and Onvansertib also confirm the bias of TransPhamer towards the target pharmacophore, with a median pharmacophoric similarity of 0.92, and demonstrates the capability of TransPhamer to explore distinct structures, with a median molecular similarity of 0.28 (Fig. 4b).

We then carried out virtual screening against the generated compound library to identify drug-like hit compounds targeting PLK1. First, molecules exhibiting pharmacophoric similarity to Ovansertib below 0.85 were eliminated. Second, Lipinski's rule of five[62] with the maximum allowed molecular weight set to 1000, and medicinal chemistry filters[45] were applied to retrieve drug-like generated molecules. Third, molecules containing the same pyrazolo-quinazoline core as Onvansertib were removed. While TransPharmer often produces distinct structures, it also tends to generate the identical moieties of the reference compound which best satisfy the conditional pharmacophore fingerprint. The remaining compounds were then docked into the ATP-binding pocket of PLK1 using Glide in standard precision mode[63]. Polar interaction (hydrogen bonds or salt bridges) between ligands and residue Lys82, Cys133, Glu140, and Asp194 were examined using PLIP[64]. Compounds with a docking score better than -9.0 kcal/mol and forming polar interaction with at least two key residues (where hinge region residue Cys133 is requisite) were kept. These molecules then underwent a two-step clustering process. First, identical Bemis-Murcko scaffolds[65] were grouped; second, the scaffolds were clustered using the Butina algorithm[66], efficiently implemented in chemfp[67], with Morgan fingerprints[47] (radius 2, 2048-bit) as molecular descriptors and a distance threshold of 0.1. 2300 representative members from each cluster with the best docking score and ligand efficiency (docking score divided by molecular weight) were selected and docked into PLK1 again using Glide in extra precision mode[68]. Upon completion of the docking, the molecules were ranked based on their overall performance, considering docking score, ligand efficiency, and the binding mode of the No.1 pose.

We systematically inspected the top-ranked generated molecules and selected 42 candidate compounds taking into account factors such as synthesizability, novelty, and the diversity of the generated compounds. The comprehensive listing of the molecular structure for these 42 compounds is available in the Supplementary Information (Supplementary Figs. 3 and 4). These compounds were classified into five groups based on their core fragments that potentially bind to the hinge region of the kinase domain of PLK1 (Fig. 4c). A detailed examination of the known PLK1 inhibitors sharing these cores revealed that the majority of them exhibit low bioactivities, with the exception of ligands featuring core 2, displaying moderate to high bioactivities. Notably, 2/3 (28 out of 42) of our generated compounds carry core 1, whereas only one known active ligand features this core. This underscores the novelty of the new scaffolds containing core 1 as potential PLK1 inhibitors. Subsequently, these 42 compounds underwent binding free energy estimation using MM/GBSA[69–72] and evaluation of binding stability through 100 ns MD simulation. Among them, four compounds were selected based on their estimated binding free

energies and consistent binding behavior within the pocket, of which three compounds carried core 1 while one compound featured core 2.

These four compounds, namely lig-3, lig-182, lig-524, and lig-886, were subjected to chemical synthesis. Several minor modifications were made to the generated structures due to the intricacies of chemical synthesis and the need to address potential metabolic instability. The finally synthesized structures largely adhered to the designed structures by TransPharmer (Fig. 4d). To clarify, these synthesized structures are referred to as IIP0942, IIP0943, IIP0944, and IIP0945, corresponding to the original lig-3, lig-182, lig-886 and lig-524, respectively. The chemical synthesis route of IIP0943 is shown in Fig. 5a; detailed chemical syntheses of all identified compounds are presented in the Supplementary Methods. The synthesis process for IIP0943 starts with the reaction of 3-methoxy-2-nitrobenzaldehyde (943-0) and methyl 2-mercaptoacetate to yield 943-1. Subsequent removal of the methyl group resulted in 943-2. The reaction of 943-2 with 2,4-dichloro-5-methylpyrimidine produced 943-3. The formation of intermediate B occurred through a Buchwald-Hartwig amination reaction between 5-bromo-2-methoxyaniline and 1-methylpiperazine. The intermediate B was then combined with 943-3 in another Buchwald-Hartwig amination reaction, leading to the formation of ester 943-4. Subsequent treatment with ammonia/methanol resulted in the production of the final compound, IIP0943.

The obtained compounds were then tested for their inhibitory activities against PLK1 kinase. Out of the four tested compounds, three show activities with half maximal inhibitory concentration (IC$_{50}$) less than 1 $\mu$M (Fig. 5b, c). Notably, IIP0943 emerges as the most potent among them, with an IC$_{50}$ value of 5.1 ± 1.7 nM against PLK1, while Onvansertib exhibits an IC$_{50}$ of 4.8 ± 0.7 nM. The confidence intervals of IC$_{50}$ values can be found in Supplementary Table 7. To investigate the selectivity of these compounds, the IC$_{50}$ values against other Plks and FAK kinase were determined for the two most potent compounds, namely IIP0943 and IIP0942. The inclusion of FAK kinase was prompted by the identification of a potent FAK inhibitor, BI-4464, which exhibits structural similarity to and forms a comparable binding pose to IIP0943 (PDB ID: 6I8Z). This similarity was revealed during our molecular novelty assessment, where we searched for analogous compounds to IIP0943 in the ChEMBL database (Section "Molecular novelty assessment" and Supplementary Fig. 8).

The results indicate that both IIP0942 and IIP0943 exhibit excellent selectivity towards PLK1 within the PLK family (Table 2). IIP0943 also shows moderate inhibition against FAK, with an IC$_{50}$ of 264 ± 32 nM, which is over 50-fold less potent than its inhibitory effect against PLK1. IIP0942 also exhibits an IC$_{50}$ of 87.4 ± 11.1 nM against FAK, with an over two-fold selectivity for PLK1. These compounds were further tested on the HCT116 cell line and IIP0943 showed cell proliferation inhibition with an IC$_{50}$ of 0.22 ± 0.003 $\mu$M (Fig. 6a and Supplementary Fig. 5).

To understand the potency and selectivity of IIP0943, IIP0943 was docked into the ATP-binding pocket of PLK1. The 4-(benzo[b]thiophen-7-yloxy)pyrimidine core of IIP0943 resides between Cys67 and Phe183 (not depicted due to space constraints). The 5-methyl group in the 2-aminopyrimidine is accommodated by a hydrophobic pocket formed by Ala80, Val114, and Leu130 (Fig. 6c). Four hydrogen bonds are formed: the 2-aminopyrimidine moiety forms two hydrogen bonds with the backbond NH and C=O groups of the hinge region Cys133; the amide group forms one hydrogen bond with the side chain of Lys82, and another hydrogen bond with Asp194 in the DFG motif. Under physiological conditions, the 4-methylpiperazino moiety becomes protonated, forming a salt bridge with Glu140. This interaction is believed to contribute to the discernible PLK1 selectivity vs PLK 2–3, since the same type of interaction is hampered in both PLK2 and PLK3 where Glu140 is replaced by histidine[53,73,74]. The superposition of the docking pose of IIP0943 and the crystal pose of Onvansertib revealed a

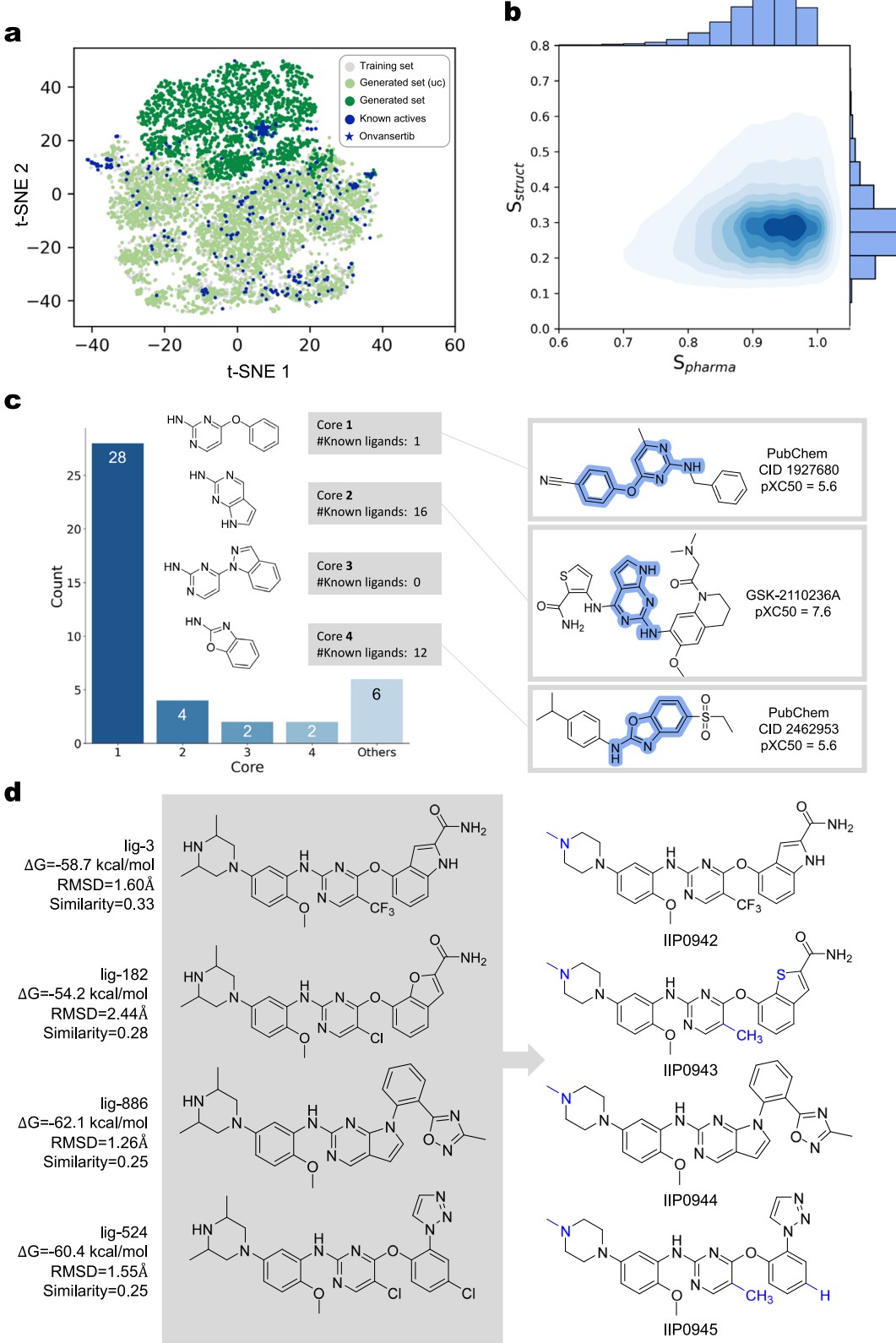

**Fig. 4 | Generation of a virtual compound library against PLK1 using Trans-Pharmer and compound prioritization. a** Illustration of the chemical space coverage of the generated compounds. T-SNE plots depict the distribution of molecules, with each 5000 randomly selected compounds from the training set, the unconditionally generated set (Generated set (uc)) the conditionally generated set, and 574 PLK1 active ligands. **b** Evaluation of pharmacophoric and molecular similarities between the generated compounds and Onvansertib. **c** Categorization of 42 candidate compounds based on the core fragment potentially binding to the hinge region of the active site. Boxes on the right: the most bioactive known PLK1 ligands that carry the substructure of the corresponding category, as well as their bioactivity recorded in the uniform expression of "pXC50" in the ExCAPE-DB database (e.g., pIC50 = 9 correspoFnds to an IC$_{50}$ value of 1 nM). The cores are highlighted in blue shade. **d** Comparison of TransPharmer's designed structures (left) and the synthesized structures with experimental validation (right). Structural modifications are highlighted in blue. The estimated binding free energy, the root-mean-square deviation (RMSD) of binding poses, and Morgan similarity scores to Onvansertib are provided for the generated compounds. Source data are provided as a Source Data file.

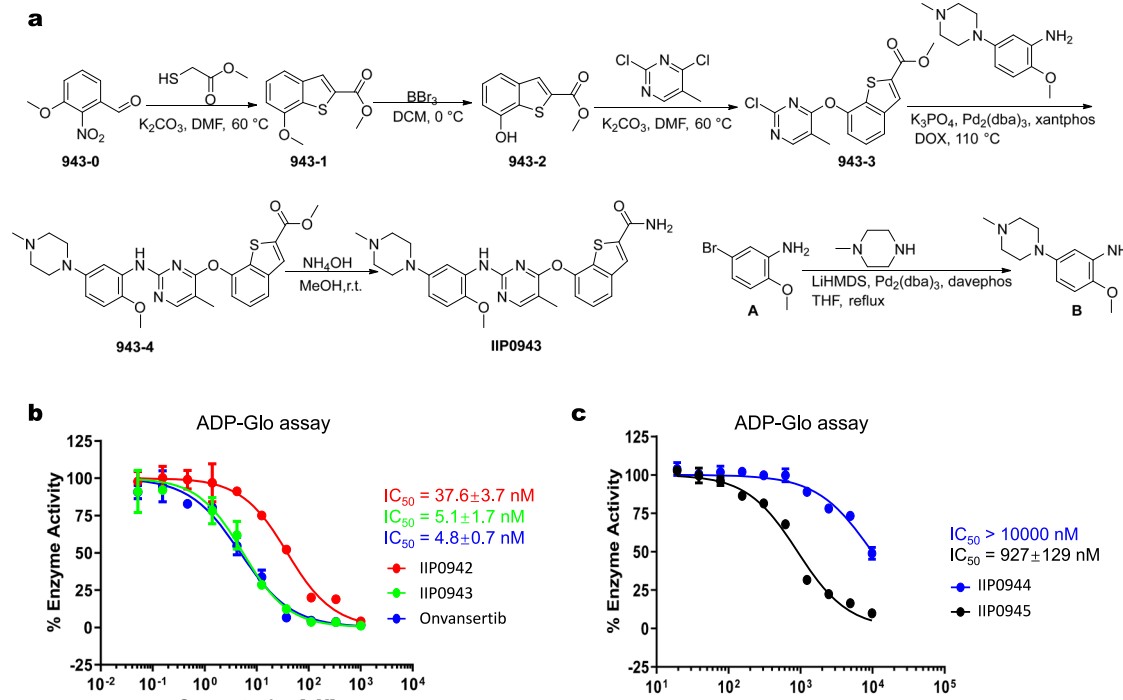

**Fig. 5 | The synthetic route and the enzymatic inhibition activities of the designed compounds. a** The chemical structure and synthetic route of IIP0943. **b** Concentration-activity curves of IIP0942, IIP0943, and Onvansertib in the PLK1 kinase ADP-Glo assays, respectively. Data are presented as mean ± standard deviation ($n = 3$ replicates). **c** Concentration-activity curves of IIP0944 and IIP0945 in the PLK1 kinase ADP-Glo assays, respectively. Data are presented as mean ± standard deviation ($n = 3$ replicates).

noteworthy distinction in the orientation of IIP0943's benzo[*b*]thiophene-2-carboxamide moiety (Fig. 6d). This moiety points towards residues in the back cleft from a different angle, which could potentially be compensated by the flexible side chain of Lys82.

Finally, we performed a comprehensive novelty assessment of the identified hit compounds. By searching for structural analogs in public databases, we confirmed the novelty of the designed PLK1 inhibitors, particularly IIP0943, across three levels: within known PLK1 active compounds, within reported bioactive ligands, and within patented molecules (refer to Supplementary Notes "Molecular novelty assessment of the discovered hits"). This evaluation highlights the capability of TransPharmer to make meaningful contributions to real-world drug discovery efforts.

## Discussion

### More on chemical space exploration
Yoshimori and colleagues discussed the distinction between structure- and pharmacophore-steered molecular generation in their reinforcement learning-based approach[75]. They compared two agent networks, each guided by rewards based on either molecular similarity or pharmacophoric similarity to a known ligand associated with the target of interest. One notable observation was that the agent guided by the molecular similarity reward successfully generated a larger number of molecules exhibiting topological similarities to the reference ligand,

but essentially failed to produce any molecules with a satisfactory pharmacophoric score. This finding implies the inherent limitations of molecular similarity-constrained methods when it comes to exploring the local chemical space.

In our study, we discovered that methods focused on generating structurally analogous compounds could yield molecules that share similarities in both topological structure and ligand pharmacophores. This finding is rational since the concept of ligand pharmacophore is rooted in molecular structure. Moreover, we made an intriguing observation that these two modes of exploration can be complementary, covering distinct regions within the local chemical space. They can also overlap when a fine-grained pharmacophoric representation is employed along with a high molecular similarity cutoff.

### Potential biases in the case study of PLK1 inhibitors design
We think the following aspects might introduce biases that could affect the current results in the case study of designing PLK1 inhibitors.

**During molecule generation.** (a) Input pharmacophore fingerprint/ reference compound. Since TransPharmer is a conditional generative model, the choice of input condition (pharmacophore fingerprint of the reference ligand) could be the largest source of bias in this work. We selected Onvansertib as the reference ligand for its potency and high selectivity towards PLK1, as well as its recent activity in clinical trials. We also used its follow-up derivatives (compounds **13** and **25**)[76] as inputs for TransPharmer to generate compounds in our in-house tests and observed slight variations in chemical space coverage (visualized by t-SNE plots). We surmise that using pharmacophore fingerprints from other unlike PLK1 inhibitors would result in significant differences in generated compounds. (b) Model hyperparameters. One key hyperparameter is the sampling temperature ($t$). This parameter re-weights the multinomial distribution of each token in generated SMILES strings, with lower temperatures increasing the probability of the top-ranked tokens relatively. In our tests, a higher

**Table 2 | Enzymatic activity of IIP0942, IIP0943, and Onvansertib against PLK1/2/3 and FAK, respectively**

| Compound | IC₅₀ (nM) | | | |
|---|---|---|---|---|
| | PLK1 | PLK2 | PLK3 | FAK |
| IIP0942 | 37.6 | >1000 | >1000 | 87.4 |
| IIP0943 | 5.1 | >1000 | >1000 | 264 |
| Onvansertib | 4.8 | >1000 | >1000 | >1000 |

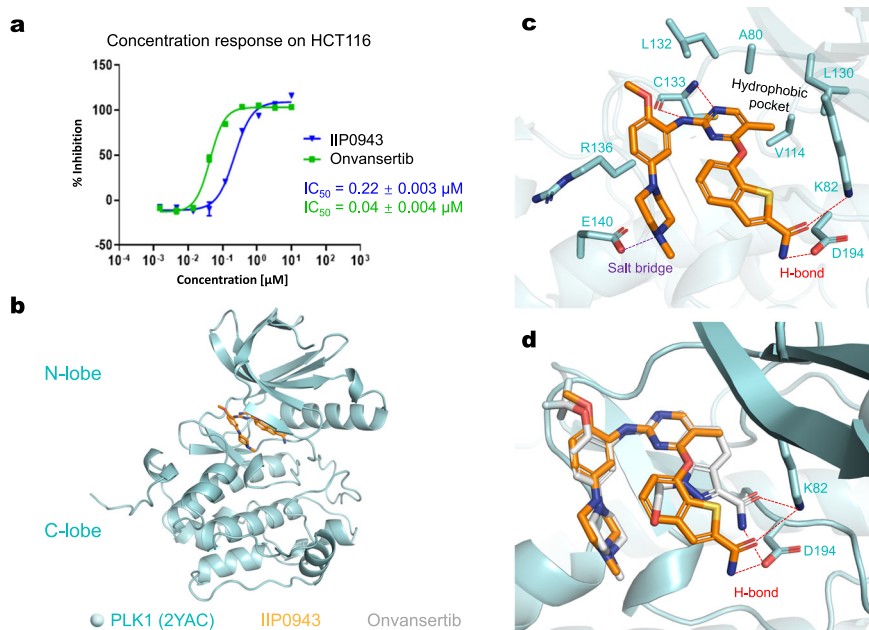

**Fig. 6 | The cellular inhibition activity and the docking pose of the generated compound IIP0943. a** The concentration-activity curves of IIP0943 and Onvansertib in the CellTiter-Glo assays on HCT116 cell lines. Data are presented as mean ± standard deviation (*n* = 3 replicates). **b** Global view of PLK1 (PDB ID: 2YAC) with the stick model of IIP0943 (in orange) docked into the active site of PLK1.

**c** Close-up view of the suggested binding pose of IIP0943 and the interactions between IIP0943 and the PLK1 kinase domain (key residues are shown as pale cyan sticks). **d** Superimposition of IIP0943 (in orange) and the co-crystallized Onvansertib (in gray) in the active site of PLK1. Hydrogen bonds in the back cleft are shown with dashed red lines.

sampling temperature (*t* = 1.2) improved diversity but also significantly reduced the performance of top-ranked compounds compared to a default lower temperature (*t* = 0.7). We expect there are some balance points to improve diversity without losing much performance. We stuck to the sampling temperature at 0.7 in our work, but users may explore this hyperparameter further in their own studies by adjusting it in the configuration file.

**During virtual screening.** (a) Novelty filters. Although TransPharmer can generate novel compounds, chemical structures that are highly similar to the reference compound also appeared in the generated set as they easily fulfilled the target pharmacophore condition. To avoid the overrepresentation of these similar compounds among top-ranked compounds, we used SMARTS patterns to retain structurally novel compounds. In the PLK1 case study, we used the pattern `c12ncncc1-CC-[n,c]3:[n,c]:[n,c]:[n,c]32` to filter out molecules with scaffolds similar to the pyrazolo-quinazoline core of Onvansertib. This novelty filter was very effective, but a different SMARTS pattern may significantly impact the results. (b) Target binding mode. Since we used Onvansertib to provide the input pharmacophore fingerprint, we focused on four polar interactions between Onvansertib and PLK1: hydrogen bonds with Cys133, Lys82, and Asp194, and a salt bridge with Glu140. Generated compounds were scored based on the occurrence of these interactions in their docked complex with PLK1, with a mandatory requirement for forming hydrogen bonds with Cys133. Different scoring criteria might yield different outcomes.

**During manual inspection.** After the virtual screening, a ranked list of 2300 generated compounds was cherrypicked to retain 42 promising compounds for further evaluation. We specifically focused on some aspects of Onvansertib during our visual inspection and cherrypicking. These may constitute potential biases in the case study of PLK1 as well. (a) Core region. One objective of the PLK1 case study was to identify compounds with distinct scaffolds from the pyrazoloquinazoline core moiety of Onvansertib, which is patented (such as WO2008074788). During the manual inspection of the generated

compounds, we prioritized novel scaffolds while tolerating those containing Onvansertib's 1-phenylpiperazine moiety. (b) 3D shape. When inspecting the docking poses, we tended to select compounds adopting a similar U-shape pose to Onvansertib (PDB ID: 2YAC), although we also considered promising molecules with different binding poses (e.g., L-shape).

## Model controllability and interpretability
We conducted three conditional tests to examine the controllability of our TransPharmer model under different conditions: (i) setting all 72-dimensional bits to 0; (ii) setting all 72-dimensional bits to 1; (iii) setting one bit to 1 while keeping the other bits at 0, thereby generating 72 single-activated-bit fingerprints. These fingerprints were then used as condition vectors to generate molecules. As expected, the outcomes of conditions (i) and (ii) were random. For condition (iii), we collected the frequency of each bit under the generation conditions. The resulting heatmap is shown in Supplementary Fig. 10. According to the results, our model can be effectively controlled by single-activated-bit conditions, producing molecules that are enriched in features corresponding to the high-frequency activated bits. The low-frequency activated bits, however, exhibited a random activation pattern, indicating that the model struggles to learn these less frequent features, which correspond to relatively rare structural motifs, such as positively or negatively charged groups, zinc-ion-binding moieties, and their combinations. This limitation could potentially be addressed by further fine-tuning with relevant datasets.

To further investigate what the models have learned, we analyzed the attention maps across all transformer blocks and attention heads. Though most attention maps appeared sparse, particularly in higher layers (Supplementary Fig. 6a), we did observe meaningful patterns in some densely activated maps. As exemplified in Supplementary Fig. 6b, where the oxygen atom of the ligand, which is the only hydrogen bond acceptor, activates all the corresponding acceptor-related bits in the pharmacophore fingerprint. However, these observations were limited and did not capture all the relationships between

each fingerprint bit and the corresponding molecular features as one might expect.

These may be caused by the following reasons: (1) The attention maps may not capture all parts of a generated compound. The linker atoms between pharmacophore features are not expected to activate attention, which could explain the sparsity of the attention maps. The knowledge required to generate proper linkers likely resides in the auto-regressive probabilistic distribution learned by the decoder, but this information cannot be revealed by attention maps. (2) The information from different positions becomes increasingly mixed in higher layers. This accounts for the distinctions observed in the attention maps in the first couple of layers, while in the higher layers, the attention weights tend to be more uniform. These observations are consistent with findings from previous studies and could potentially be addressed by techniques such as attention rollout or attention flow[77]. However, currently, there are no universal methods for analyzing attention maps. Even in well-defined natural language processing tasks, a technique that works well for one task may fail for another[77]. Therefore, fully understanding how generative models learn requires further study.

### On 3D pharmacophores

The complementary nature of protein-ligand interactions as 3D spatial pharmacophores is widely recognized. Ligand-based pharmacophores are analogous to a particular kind of negative image in the binding site. The generated compounds by our model may satiate the actual 3D binding pharmacophores given a predetermined 2D pharmacophore fingerprint. Actually, by listing all potential pharmacophore topologies in the measured Euclidean distances of two or three points, the 3D spatial pharmacophores may be transformed into a variety of 2D pharmacophore topologies. We think that it is doable to manually design a desirable pharmacophore topology. As a result, it is simple to discretize a set of appropriate pharmacophore topologies into bit fingerprints that are numerous criteria to steer molecular generation. Even while ligand-based 3D pharmacophores offer one option to guide the generative model, it is still challenging to guarantee that active ligand conformations are generated. To get around this issue, one might predict active ligand conformations using other deep-learning models[78].

### Toward more universal generative models

The pharmacophoric fingerprints utilized in TransPharmer serve as valuable prompts, enabling the model to seamlessly transition between designing ligands for different targets without requiring additional fine-tuning. This capability was demonstrated in the case studies on DRD2 and PLK1, showing that TransPharmer can be readily applied in diverse scenarios.

Compared to other molecular generative models based on GPT-like architectures[18], TransPharmer offers two primary contributions. By prompting with pharmacophoric fingerprints, TransPharmer incorporates prior knowledge into the generation of pharmaceutically relevant compounds, thereby aligning more closely with the goals of medicinal chemists. This approach also paves the way to the development of extensive pharmaceutical generative models that integrate multimodal knowledge alongside basic chemical principles derived from molecular structures[79]. Additionally, TransPharmer leverages the structural hopping properties of pharmacophores to aid in discovering novel compounds with bioactivity against the same pharmaceutical targets.

Nonetheless, several directions can be explored in the future to enhance the model's versatility and general applicability. First, additional generation modes, such as fragment-linking, should be incorporated alongside de novo generation and scaffold elaboration. Advances in unordered chemical language modeling can directly support these functionalities[80]. Second, generative models that

produce easily synthesizable molecules are preferable, as they can accelerate the timeline for wet lab experimental validation[81]. Finally, multi-objective optimization should be integrated into the generative process to support more efficient design. Recent advances, such as integrating Pareto optimization with generative models, may help identify novel compounds with a balanced profile[82].

## Methods

### Pharmacophore features and fingerprint extraction

The molecular graph is first converted into a fully connected graph of pharmacophore features using the definition of ligand-based pharmacophores from RDKit v2021.9 (*BaseFeature. fdef*)[83]. This definition encompasses eight types of pharmacophore features, including hydrogen-bond acceptors and donors, aromatic rings, moieties possessing positive or negative ionizability, hydrophobic entities, or those associated with Zn ion binding. Detailed patterns for each type are presented in Supplementary Table 8. To derive the pharmacophore fingerprints utilized in TransPharmer, we obtained two-point and three-point combinations of pharmacophore features, as well as the shortest topological distances between each feature pair. The topological distances were discretized into 2-bin (the range for short distances as [0, 3) and for long distances as [3, 8)) or 3-bin (the range for short distances as (0, 2), for medium distances as [2, 5) and for long distances as [5, 8)) signals. When a distance falls within a specific range, the corresponding bit is set to 1, otherwise 0; if the distance exceeds the maximum considered distance, 8 in this study, there will be null signals (00 or 000). For the two-point pharmacophoric features with 2-bin and 3-bin discretization schemes, the lengths of the binary pharmacophore fingerprints obtained are 72 and 108, respectively. For the combination of two- and three-point pharmacophoric features with the 2-bin scheme, the lengths of the fingerprints are 1032. The fingerprint extraction process was built based on the 2D pharmacophore fingerprint modules implemented in RDKit[48].

### Model architecture

As illustrated in Fig. 1 and Supplementary Fig. 11, TransPharmer receives the pairings of a SMILES string and its extracted pharmacophore fingerprint as two-channel input during training. After segmental encoding and positional encoding, these input embeddings are fed into the Transformer decoder with multi-head self-and-mixed attention blocks to decode the SMILES tokens in the next position. The segmental encoding aims to distinguish between tokens and conditions by using explicit label vectors (0s for pharmacophore fingerprint and 1s for SMILES token). The positional encoding adopts a rotary positional encoding[84] by multiplying the embedding vectors by the rotation matrix as follows,

$$Attention(Q, K, V)_m = \frac{\sum_{n=1}^{N} (R_{\Theta,m}^d \phi(q_m))^T (R_{\Theta,n}^d \varphi(k_n)) v_n}{\sum_{n=1}^{N} \phi(q_m)^T \varphi(k_n)} \quad (1)$$

where $\varphi(*)$ and $\phi(*)$ are usually non-negative functions, $R_{\Theta,m}^d$ and $R_{\Theta,n}^d$ are rotation matrix. This positional encoding was demonstrated to be more compatible with the linear operation in the attention block and to converge faster during training.[84] A slim version of the GPT-3 model[42] is utilized for the multi-head Transformer decoder. And self-and-mixed attention blocks are adopted for adequate information exchange in order to learn implicit associations. With the processing of the Transformer decoder, the final layer outputs the probabilities of the next SMILES tokens using linear transformation and softmax operations. The hyperparameters of TransPharmer are shown in Supplementary Table 9.

### Data set setup

We use the GuacaMol dataset[44], which is derived from the ChEMBL24 database and is composed of about 1.6 million unique

compounds. The sizes of the training, validation, and testing sets are 1,273,104 (80%), 79,562 (5%), and 238,681 (15%), respectively, for model development and evaluation, following the data splitting of GuacaMol. All TransPharmer models in the pharmacophore-constrained molecule generation tasks were trained on the Guaca-Mol dataset.

A 108-token vocabulary was first constructed from the SMILES strings from the GuacaMol dataset, which contains '#', '%10', '%11', '%12', '(', ')', '-', '1', '2', '3', '4', '5', '6', '7', '8', '9', '<', '=', 'B', 'Br', 'C', 'Cl', 'F', 'I', 'N', 'O', 'P', 'S', '[B-]', '[BH-]', '[BH2-]', '[BH3-]', '[B]', '[Br-]', '[Br+2]', '[C+]', '[C-]', '[CH+]', '[CH-]', '[CH2+]', '[CH2]', '[CH]', '[Cl+]', '[Cl-]', '[Cl+3]', '[Cl+2]', '[F-]', '[F+]', '[H]', '[I+]', '[I+2]', '[I+3]', '[IH2]', '[IH]', '[I-]', '[N+]', '[N-]', '[NH+]', '[NH-]', '[NH2+]', '[NH3+]', '[N]', '[O+]', '[O-]', '[OH+]', '[O]', '[P-]', '[P+]', '[PH+]', '[PH2+]', '[PH]', '[S+]', '[S-]', '[SH+]', '[SH]', '[Se-]', '[Se+]', '[SeH+]', '[SeH]', '[Se]', '[SeH2]', '[Si-]', '[SiH-]', '[SiH2]', '[SiH]', '[Si]', '[SH-]', '[b-]', '[bH-]', '[c+]', '[c-]', '[cH+]', '[cH-]', '[n+]', '[n-]', '[nH+]', '[nH]', '[o+]', '[s+]', '[sH+]', '[se+]', '[se]', 'b', 'c', 'n', 'o', 'p' and 's'. After removing less frequent tokens (including '[Br+2]','[Br-]','[Cl+2]','[Cl+3]','[Cl+]','[Cl-]','[F-]','[I+2]','[I+3]','[I-]','[P-]','[SH-]','[Se-]','[SeH2]'), a 94-token vocabulary is used to process SMILES strings from different sources. Those containing tokens outside the vocabulary were removed.

8323 DRD2 actives were collected from the ExCAPE-DB[85] and 7939 were left after the elimination of invalid SMILES strings (can not parsed by RDKit) and duplicate structures (share the same canonical SMILES strings). Over 40,000 DRD2 inactives were also downloaded from the ExCAPE-DB and 7939 of them were randomly sampled for visualization and comparison with actives. TransPharmer in the recall experiment of DRD2 actives was retrained on the merged dataset of GuacaMol and DRD2 actives, described in Section "Settings for DRD2 recall experiment".

3873 entries of PLK1 actives were also collected from the ExCAPE-DB database and all of them have valid and non-duplicate SMILES strings. Each entry contains the molecular structure in SMILES format and the bioactivity record in the uniform expression of "pXC50" (e.g., pIC50 or pEC50. pIC50 = 9 corresponds to an $IC_{50}$ value of 1 nM).

### Settings for pharmacophore-constrained molecule generation
Three-hundred compounds (referred to as conditioning compounds) were randomly selected from the reserved test set, and each model used their pharmacophoric information to guide the de novo generation of novel molecules. For the task of scaffold elaboration, each conditioning compound is fragmented into two parts by breaking a random acyclic single bond between two non-hydrogen atoms. One fragment of the conditioning compound is chosen arbitrarily as the core or starting fragment, while the other becomes a reference elaboration. Using the core fragments as starting points, each model performs scaffold elaboration guided by the pharmacophoric information of the reference fragments. For both tasks, each model attempts to generate 600 molecules for every conditioning compound, and invalid and duplicate molecules are filtered out before further evaluation. Detailed parameter settings for the three external baseline models can be found in the Section "The settings of compared methods".

### The settings of compared methods
**LigDream[35].** LigDream can generate novel molecules guided by the three-dimensional shape and pharmacophoric features of a reference compound. LigDream contains a shape variational auto-encoder (VAE), which encodes a voxelized 3D molecular structure into its latent code and reconstructs the voxelized compound representation from it, and a shape captioning recurrent neural network (RNN), which decodes the voxelized representation to the SMILES of a specific molecule. The LigDream authors found that the VAE reparametrization factor $\lambda$ and the RNN probabilistic sampling

can provide different sources of sampling variability. In this study, we set the reparametrization factor $\lambda$ to 1.0 and turned off the RNN probabilistic sampling as suggested. The model weight was obtained from their public repository (https://github.com/compsciencelab/ligdream).

**PGMG[39].** PGMG receives a fully connected graph containing selected pharmacophore features. This graph is encoded using a Gated Graph Convolutional Network to obtain an embedding vector, which is subsequently decoded into SMILES strings using transformer encoder-decoder blocks. In this study, following the training process of PGMG, a pharmacophore hypothesis was constructed by randomly selecting 3–7 pharmacophore features for each test case molecule, and the 3D coordinates for each feature were obtained from the molecular conformation embedded using the ETKDG[86] method, as implemented in RDKit. We utilized the pretrained PGMG (accessible at https://github.com/CSUBioGroup/PGMG) to generate 600 samples for each pharmacophore hypothesis. For other exploration settings and evaluation of PGMG, please refer to Table S2 in the supplementary materials.

**DEVELOP[34].** DEVELOP integrated pharmacophoric information of the regions to be explored into the process of fragment linking or scaffold elaboration and has shown broad potential in scenarios such as PROTAC design or R-group optimization. We utilized the scripts provided by the authors of DEVELOP to prepare the pharmcophore information and index files and perform the required preprocessing for our testing data. We loaded the pretrained model weights (accessible at https://github.com/oxpig/DEVELOP) and adopted the default parameters during generation following the instructions of the setting used to generate molecules with the same number of atoms as the reference molecule.

**STONED[51].** Superfast traversal, optimization, novelty, exploration, and discovery (STONED) is an algorithm that can perform local chemical subspace exploration around a target molecule and other functionalities. STONED achieves these by modifying the SELFIES[21] string representation of the reference molecules. The amount and location of the modified characters have different effects on the similarity between the resultant structures and the original ones. Basically, restricting the amount or the location of the SELFIES changes to either the initial or the terminal region yields similar mutated structures[51].

In this study, we utilized this feature of STONED to explore the local chemical space around a target molecule in two different modes. The first one is the default mode ("STONED") which allows the mutation positions to be chosen randomly and the number of mutations up to 5. The "STONED" mode produces both similar and dissimilar structures to the starting molecule. The second is the "STONED-focused" mode that allows only 1 modification and restricts the mutation position to the terminal 10% of the SELFIES. The "STONED-focused" mode was set up intentionally to produce highly similar mutated structures to the target one. For the "STONED" mode, we sampled 10,000 times for each molecule, while for the "STONED-focused" mode we sampled 100,000 times because the mutated structures have a high probability of repeating themselves. Only the non-duplicate parts of the generated molecules were retained by examining their canonical SMILES strings output by RDKit[48].

### Evaluation metrics
In this study, we use $S_{struct}$ to represent the molecular similarity score which is measured by the Tanimoto coefficient of 2048-bit Morgan circular fingerprints with a radius of 2, and use $S_{pharma}$ to represent the pharmacophoric similarity score which is measured by the Tanimoto coefficient of ErG fingerprints[32] implemented by RDKit. Molecules will go through charge neutralization before similarity scoring.

The deviation of pharmacophoric feature counts $D_{count}$ is formulated as follows:

$$D_{count} = \frac{1}{N}\sum_{i}^{N}\sum_{j}^{m}|n_j^i - n_j^{ref}| \qquad (2)$$

where $n_j^i$ is the number of $j$th pharmacophoric features in $i$th generated molecule, $n_j^{ref}$ is the number of $j$th pharmacophoric feature in the reference molecule, $m$ is the total number of pharmacophoric feature types ($m = 8$ in this study) and $N$ is the total number of generated molecules.

The recall rate in the recall experiment of DRD2 actives is formulated as follows:

$$Recall = \frac{\#Known\ actives\ in\ the\ generated\ set}{\#Known\ actives\ in\ the\ reserved\ set} \qquad (3)$$

The apparent precision is the number of known actives unseen by TransPharmer found within the set of 4000 generated molecules. These molecules were sampled more than once during the generation process and continuously joined the generated set until the size of 4000 was reached.

### Settings for DRD2 recall experiment
The Bemis-Murcko scaffolds of the 7939 DRD2 actives were extracted and clustered using Butina algorithm[66] implemented in RDKit[48], with Morgan fingerprints[47] (radius 2, 2048-bit) as molecular descriptors and a distance threshold of 0.4. Then, scaffold clusters were sorted by size in descending order and 3717 ligands with scaffolds in the odd-indexed clusters were added into the training set of TransPharmer, while 4222 ligands with scaffolds in the even-indexed clusters were actives to be recalled. During generation, active ligands in the training set were encoded into 72-bit pharmacophoric fingerprints and used as prompts of TransPharmer to generate 1000 SMILES per condition, yielding a total of 3,717,000 generated samples. 3717 unrelated molecules to DRD2 randomly drawn from the training set (Fig. 3b) were also encoded into 72-bit pharmacophoric fingerprints for TransPharmer to generate 1000 SMILES per condition.

### DRD2 QSAR model
A classification model employing a Support Vector Machine (SVM) for the prediction of bioactivity was developed following DeepDrugCoder[54]. The standard implementation of SVM from the scikit-learn v0.20.347 Python package was used, with the radial basis function as the kernel function. The model was trained to discriminate active compounds from inactive ones based on their 2048-bit-radius 2 Morgan fingerprint representations. Model weights and optimized hyperparameters were loaded from https://github.com/pcko1/Deep-Drug-Coder/tree/master/models. The model outputs the probability of a compound being active against DRD2.

### t-distributed stochastic neighbor embedding (t-SNE)
To visualize the chemical space encompassed by the generated molecules, the training set, and the known PLK1 active ligands, we constructed t-SNE plots. The 108-bit pharmacophore fingerprints were used as the molecular descriptors. The perplexity parameter was set to 50. A subset of known PLK1 active ligands was constructed by removing ligands with a pXC50 value lower than 6. Employing the Barnes-Hut implementation of the t-SNE algorithm[87], we obtained two-dimensional representations for 574 PLK1 active ligands and each 5,000 randomly selected molecules from the training set and two generated sets (both conditionally and unconditionally).

### Molecular docking
The receptor structure was taken from the Protein Data Bank (PDB)[88] (PDB ID: 2YAC) and prepared using the Schrodinger Protein Preparation Wizard[89] with default parameters, i.e., we added hydrogens, protonated non-residue molecules at pH $7 \pm 2$ using Epik[90], removed waters, ions and crystallization artifacts (e.g., tartaric acid), optimized hydrogen bond assignment at pH 7 using PROPKA[91] and minimized the structure using the OPLS3e force field[92]. A grid was defined using the centroid of the co-crystallized ligand Ovansertib as the center. Before the docking procedure, the generated ligands were prepared using LigPrep[93] to enumerate unspecified stereocentres, tautomers, and protonation states and perform minimization using the OPLS3e force field. Each molecule along with any respective variants was then docked using Glide[63]. We performed a redocking of Onvansertib into the ATP pocket of PLK1 to validate our docking protocol. Onvansertib was favorably scored by Glide with docking scores lower than −11 kcal/mol and the RMSD of its No.1 docking pose and the co-crystal pose is less than 0.5 Å.

### Molecular dynamics (MD) simulation
MD simulation was carried out on the systems of PLK1 in complex with generated ligands. The systems were first minimized through steepest descent minimization until the termination condition, i.e., the maximum force below 10.0 kJ/mol, was satisfied. After minimization, the systems were heated to 300 K over 100 picoseconds (ps) using the NVT ensemble with a restraint of 1000 kJ/mol nm$^{-2}$ on both the kinase and ligands, followed by an additional 100 ps of NVT equilibration with a restraint solely on the protein. Next, 100 ps of NPT equilibration was conducted. Finally, either a 4-nanosecond (ns) production run for estimating binding free energy or a 100 ns run for evaluating binding stability was conducted. The long-range electrostatics were accounted for by means of the particle mesh Ewald (PME) method, with a cutoff of 12 Å applied uniformly across all the MD simulations. All hydrogen-heavy atom bonds were constrained by the LINCS method, and simulations were executed with a time step of 2 femtoseconds. Temperature coupling utilized the V-rescale method. To assess the stability of the simulated systems, the root-mean-square deviation (RMSD) was computed based on the last 20 ns of the trajectory after performing the alignment of protein structures. We validated our MD simulation protocol by carrying out a 100 ns run for the system of PLK1 in a complex with Onvansertib. The last 20 ns RMSD is 1.71 Å, indicating high binding stability of Onvansertib in the ATP pocket of PLK1.

### Molecular mechanics with generalized Born and surface area solvation (MM/GBSA)
The MM/GBSA calculations were conducted employing gmx_MMPBSA v1.6[94], a tool derived from AMBER's MMPBSA.py. The GBOBC2 (igb = 5) model was utilized in this study, with a salt concentration set at 0.15 M. For the kinase, the ff14SB force field was employed, while the General Amber Force Field was applied to the generated ligands. Other default parameters for MM/GBSA calculations were applied.

### Chemical synthesis
We assessed the synthesizability using the Synthetic Accessibility (SA) score[95] to estimate the ease of synthesis of the designed compounds, supplemented by manual inspection by medicinal chemistry experts. The primary synthetic data are available in the Supplementary Methods.

### In vitro kinase activity assays
In vitro kinase activity assays were conducted through ADP-Glo assay services provided by Conradbio (Conradbio, China). The protocol for the PLK1 assay is described as follows (protocols for other kinases are very similar). Enzyme, substrate, ATP, and compounds were diluted in a Kinase Buffer composed of 40 mM Tris (pH 7.5), 20 mM $MgCl_2$,

0.1 mg/ml BSA, and 50 $\mu$M DTT. In a 384-well low-volume plate, 1 $\mu$l of the compound or 5% dimethyl sulfoxide (DMSO), 2 $\mu$l of PLK1 enzyme (15 ng/well), and 2 $\mu$l of substrate/ATP mix (final concentration: 20 $\mu$M ATP, 0.2 $\mu g/\mu$l Casein) were added to each well. The plate was then incubated at 25 °C for 60 min to allow for kinase activity. Following the enzymatic reaction, 5 $\mu$l of ADP-Glo™ Reagent was added to each well, and the plate was incubated at 25 °C for an additional 40 min. Subsequently, 10 $\mu$l of Kinase Detection Reagent was added, and the plate was incubated for a final 30 min at 25 °C. Luminescence was recorded with an integration time of 0.5 s.

IC50 values were calculated using Prism 8 by fitting the following equation:

$$Y = Bottom + (Top - Bottom)/(1 + 10^{(\log IC_{50} - X)} \times HillSlope), \quad (4)$$

where $X$ is a log of concentration, $Y$ is a response, and top and bottom are the responses of controls. Each assay was repeated at least three times, and we computed the mean and standard deviation for the values.

## Cell viability assays
Cell viability assays were conducted through CellTiter-Glo assay services provided by Conradbio (Conradbio, China). The protocol is briefly described as follows. Firstly, when the cell confluence reaches 80%, cells are collected and counted. Subsequently, a cell suspension is diluted, and 80 $\mu$l of the suspension is seeded into each well of a 96-well U-bottom plate. The plate is then placed in a 37 °C, 5% $CO_2$ incubator for optimal cell growth. After 24 h of incubation, a 20 $\mu$l aliquot of a diluted compound solution is added to specific wells on the plate, 0.5% dimethyl sulfoxide (DMSO) is used as a negative control. Following the compound addition, the plate is returned to the incubator for an additional day. Upon completion of the incubation period, the CellTiter-Glo assay (Promega) is performed according to the manufacturer's manual. This assay is designed to measure cell viability based on luminescence, providing insights into the impact of the compounds on cellular health. Finally, data calculation is carried out to analyze the results of the CTG assay using Prism 8.

## Molecular novelty assessment
A molecular novelty assessment of the designed compounds exhibiting IC$_{50}$ below 1 $\mu$M, namely IIP0942, IIP0943, and IIP0945, was performed within the ExCAPE-DB and ChEMBL databases and using SciFinder. The settings and results can be found in the Supplementary Notes.

## Reporting summary
Further information on research design is available in the Nature Portfolio Reporting Summary linked to this article.

# Data availability
Source data are provided with this paper as a Source Data file. Supplementary data in this study are provided in the Supplementary Information. The generated molecules in the case studies of DRD2 and PLK1 in this study have been deposited at Zenodo via https://doi.org/10.5281/zenodo.14227821[96]. The crystal structure of PLK1 used in this study is available in the RCSB PDB database under accession code 2YAC [https://doi.org/10.2210/pdb2YAC/pdb]. Source data are provided with this paper.

# Code availability
The source codes of TransPharmer is available at https://github.com/iipharma/transpharmer-repo and Zenodo (https://doi.org/10.5281/zenodo.14228119)[97].

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

## Acknowledgements

We thank Dr. Guangwei He for insightful discussions on experimental design and are grateful to his team at HIPI for their support during experiments. We also acknowledge Dr. Chu, Dr. Xu, and Minghan He for their assistance with statistical analysis and design. This work was supported in part by the National Key R&D Program of China (grant 2023YFF1205103), the National Natural Science Foundation of China (grants 220330010), and the Chinese Academy of Medical Sciences (grant 2021-I2M-5-014). Additionally, this work received support from Anhui's Plans for Major Provincial Science & Technology Projects (Grant 202303a07020009).

## Author contributions

Y.X. devised the ideas. J.Z. and W.X. implemented the deep learning model and performed the model training. W.X., J.Z., and Y.X. discussed and analyzed the data. W.X., Q.X., and C.G. conducted the compound screening and prioritization. Y.R., J.X., and Q.S. performed chemical structure characterization and analyzed data. Y.X., L.L., and J.P. supervised the project. W.X., J.Z., Y.X., L.L., and J.P. wrote the manuscript.

## Competing interests

The authors declare no competing interests.
