## [Transparent Peer Review file · Nature Communications]

Accelerating Discovery of Bioactive Ligands with Pharmacophore Informed Generative Models

Corresponding Author: Professor Jianfeng Pei

Version 0:

Reviewer comments:

Reviewer #1

(Remarks to the Author)

The authors propose a new pharmacophore fingerprint constrained molecular de novo design method, which they are also experimentally validating.

There are several pharmacophore based molecular de novo generators out there, but this one has innovative aspects that makes the manuscript potentially suitable for publication.

However, the comparison with other methods is insufficient, since the other methods are using 3D pharmacophores and not pharmacophore fingerprints. It would be more relevant to do a comparison with the same pharmacophore fingerprint used here implemented in a platform like REINVENT.

I don't agree that the new PLK1 inhibitors are especially novel. The left side and the hinge binding region is the same as Onvensartib. I would expect larger differences between the new compounds and onvensartib to think this would be an interesting example.

As is clear from EXCAPE-DB, there exists already several thousand PLK1 inhibitors so there is no scientific novelty in publishing more.

Also I don't see much point in using a pharmacophore finger print for finding novel PLK1 inhibitors, a docking protocol to score the generated molecules would be more suitable.

In conclusion the paper is recommended to be submitted to a more technical journal than Nature Communications.

(Remarks on code availability)

Reviewer #2

(Remarks to the Author)

The authors have designed a novel method to generate molecules conditioned on 2D pharmacophores. To generate the datasets they extract pharmacophore features using functions from RDKit and define the inter-feature distance based on the topological distance between them. The model itself is a sequence based transformer that takes pharmacophore fingerprints as conditional input. The authors show that the model generates molecules that are well conditioned on the pharmacophore input and can be a useful tool for real world discovery. They further provide two case studies on the DRD2 and PLK1 receptor systems to showcase its utility in drug discovery projects. I believe the proposed method is impactful and would be valuable to readers of this domain and hence should be published. I have, however, a few comments that may improve the quality of the publication:

Minor Comments:

The comparison with PGMG in Supplementary Table 1 and 2 is confusing:

The authors on page 6-7 mention that the TransPharmer closely approximates the match scores of PGMG but in Supplementary Table 1 it is very clear that PGMG does much better than TransPharmer, I would not say the numbers are a "close approximation" of each other.

The authors need to describe the experiments in supplementary 2 in more detail. Why were 114 molecules selected? Why is it important to evaluate molecules with 2 distinct conformers for this experiment? Are the other rows in the table on the entire test set? Furthermore, why are some of the average deviation numbers negative? Should deviation not always be a positive value? Finally, what's the final takeaway message from these results? - The authors should add another paragraph or 2 explaining this more.

I believe the paper's quality will also be improved if they did an unconditional generation benchmark on the MOSES dataset as well, as there are many 2D generation model that benchmark on that dataset and it would provide a nice understanding on the molecule generation capability of the model

Is the precision numbers in Supplementary Table 6 based on only 4000 molecules? It would be useful to mention that explicitly in the table caption.

Page 12 - It would be useful to expand on DeepDrugCoder, is the predictive model same as the QSAR model? And what is the reserved set mentioned here?

Finally, there is a lot of useful content in the supplementary information that is not referenced in the main text, it may help the reader get more context about the method if those experiments are discussed briefly/ referenced in the main text.

(Remarks on code availability)

The code is not accessible

Reviewer #3

(Remarks to the Author)

In the manuscript, "Accelerating Discovery of Novel and Bioactive Ligands With Pharmacophore-Informed Generative Models" by Xie et al., the authors describe the development of an innovative generative model to identify not only novel but also bioactive compounds. They described a 2D pharmacophore-based fingerprint approach for the generation of structurally diverse but pharmacophorically similar molecular structures. Their aim was to provide suggestions to explore the active chemical space for a target and achieving a high probability of activity for the generated compounds. The approach is innovative and makes use of one of the main advantages of pharmacophore-based similarity searches: scaffold hopping. In general, a very interesting study.

While the benchmarking experiments and retrospective study on DRD2 ligands showed promising data but also already suggest some of the challenges of this approach, the experimental case study somehow confirms some limitations of this method in real life projects. Actually, the case study is the main reason why I think this work is exceptional and very informative: In their case study on PLK1 inhibitors, TransPharmer was used to generate novel structures as suggestions for synthesis. The model returned numerous structures, however, many of them are from the same scaffold as training compounds, includes duplicates, non-drug-like structures etc. Using additional filters, the suggested structures could be narrowed down to the most promising ones.

The virtual screening part included two docking runs and MD simulations. Unfortunately, the validation of this workflows has not been described.

The 42 top-ranked molecules were further inspected and the final selection of compounds for synthesis was based on synthesizability, novelty, and the diversity of the generated compounds. Anyways, the finally synthesized compounds slightly differed from the computed structures. Notably, the 2,6-dimethylpiperazine moiety was replaced by 1-methylpiperazine, which is also found in onvansertib, making it more similar to known active compounds.

The authors further included an enzyme activity as well as a proliferation assay in the human colon cancer cell line HCT116 to verify the computationally generated hits.

In the end, the team succeeded in the development of a highly active, structurally novel compound and two other compounds with moderate activity.

The reviewers suggest the following additions and corrections:

- In general, we would recommend rereading the whole manuscript concerning spelling mis-takes, gaps between numbers and the corresponding unit and grammar guides especially in the methods section.
- describe the validation of the docking and MD runs
- please describe how synthesizability was assessed
- please include C13 NMR spectra of the newly synthesized compounds in the supporting information
- how did the actually synthesized compounds perform in the virtual screening?
- clearly address points of bias in the case study on PLK1 inhibitors and how they may have influenced the results

- in all graphs and the text, replace dose-response with concentration-response. Dose refers to e.g. mg/kg and is applicable to in vivo data only.
- concerning the enzyme activity and cell proliferation assay, we recommend presenting the IC50 values in a separate table in order to ensure clarity. Please explain how the IC50 values were calculated. Which statistics were used for enzyme activity as well as cell proliferation assay? Please add the exact method behind.
- Please show the confidence interval (CI) of the IC50 values instead of the standard deviation or as additional information, since the CI in this case is much more significant.
- Some measurement points in Figure 5b, Figure 6a and Supplementary Figure 4 have error bars, whereas others do not. Especially for cell-based assays, this is impressive. However, we're not sure, whether this is really the case. Please check again the correct display of the error bars.
- It would also be interesting to include the IIP0942 compound in the cell proliferation assay conducted in HCT116 cells. Although the substance seems not to be as effective in inhibiting PLK1 enzyme activity as IIP0943, it would be great to see the impact of the compound in a physiological context. One may also add volasertib as another PLK1 inhibitor with much higher protein affinity than onvansertib for comparison reasons.
- The protocol for in vitro kinase activity assays and cell viability assays is traceable. However, were the cells really seeded into U-bottom plates? Normally, that type of multi-well plates is used for the generation of 3D spheroids instead of monolayer cultivation.
- It would also be useful to add the effect of the DMSO control compared to the compounds in the supplement. As stated above, please add the underlying statistical analysis in the methods section for the IC50 value calculation.

(Remarks on code availability)

Reviewer #4

(Remarks to the Author)

(Remarks on code availability)

Version 1:

Reviewer comments:

Reviewer #1

(Remarks to the Author)

The authors haven't addressed any of my objections in a satisfactory way, I struggle to see the importance of the work to justify publication in Nature Communications.

(Remarks on code availability)

Reviewer #2

(Remarks to the Author)

No more comments

(Remarks on code availability)

Reviewer #3

(Remarks to the Author)

The authors have addressed the comments accordingly. In its current form, it can be published.

(Remarks on code availability)

Reviewer #4

(Remarks to the Author)

(Remarks on code availability)

The code includes all necessary information about installing and running the TransPharmer application.

Reviewer #5

(Remarks to the Author)

In this study, the authors proposed TransPharmer, a de novo molecule generation model that integrates pharmacophore fingerprints with the generative pre-training transformer. From a purely methodological/technical perspective, novelty is limited; however, the execution and the molecular analyses are quite good. I also evaluated the reviewer reports for the first round and the authors' responses. Below, I list my specific concerns.

1)

The manuscript lacks molecular novelty-based analyses. It is important to generate de novo molecules with similar pharmacophore-centric features to known effective inhibitors of the selected target for functionality; however, another important feature of generative AI models is the novelty (not only against the guiding molecules but also against the entire known chemical space). This is important to assess the model's (or we can say the proposed approach's) usefulness when utilised in real-life drug design/development pipelines. I'm not purely talking about the MOSES/GuacaMol benchmarking metric called novelty; instead, I'm talking about assessing the novel features of molecules in general (both qualitatively and quantitatively).

2)

There is no ablation study. It is important to assess which design choices led to the obtained results. One obvious comparison would be between TransPharmer and a model with the exact same ML architecture that does not utilise the pharmacophore fingerprints.

3)

In today's AI research, the interpretability of generative models is an important topic. The authors may consider an explainability-centric analysis to observe and understand what the model has learned in the context of molecular properties (e.g., using attention maps). The current state of the section entitled "Controllability of TransPharmer" does not qualify for this, but additional evaluations and discussions can solve this.

4)

It looks like there is something wrong with the t-SNE output in Figure 4a; 10,000 random training molecules and 3,873 PLK1 active ligands occupy nearly the exact same molecular space, whereas PLK1 active ligands should have occupied a specific sub-region of the random molecule space. As a result, comments made about the space occupied by the generated compounds are not valid as well. This is probably related to the hyperparameters selected for the t-SNE projection. This analysis should be repeated with different parameters to obtain a more meaningful result, and the corresponding discussion should be updated accordingly.

(Remarks on code availability)

Version 2:

Reviewer comments:

Reviewer #1

(Remarks to the Author)

The authors has done enough to convince me that the study is publishable

(Remarks on code availability)

Reviewer #5

(Remarks to the Author)

I thank the authors for their efforts. However, my concerns were only partially addressed with this revision. I'm providing detailed information regarding the missing bits below (I'm using the same numbering for the issues in my original review, considering reviewer #5).

1)

I could not find the newly added text and figure about the novelty analysis regarding the nearest neighbour similarity of the generated compounds to known DRD2 actives, neither in the main text nor the supplementary material of the revised manuscript. Where is it? It should certainly be included in the manuscript. Moreover, the information regarding this new novelty analysis in the response to the reviewers' document contains no discussion or insight. It only states, "43% of the compounds exhibit a similarity score of less than 0.4". This is insufficient. Please make the necessary modifications and

additions.

2)
An ablation study has been added to the supplementary material but has not been mentioned in the main text. This mention, together with a summary of the results obtained from the ablation study, should be added to a relevant section in the main text.

3)
Similar to issue number 1, the analyses conducted to address this issue have not been included anywhere, not in the main text or the supplementary material. Even though the results of the analysis were inconclusive, the findings should be reported and discussed in the manuscript. It is important to discuss the possible reasons behind not being able to identify clear qualitative relationships between individual pharmacophore fingerprint bits and specific molecular moieties. By referring to the figure, it would be good to mention and discuss those densely activated maps in a few attention heads within the middle layers.

Finally, a future work paragraph should be added to the conclusion section, in which information regarding “more detailed work is to be conducted in this scope” should be mentioned. Other possible future work should also be discussed in that paragraph as well.

4)
This one has been addressed.

(Remarks on code availability)

Version 3:

Reviewer comments:

Reviewer #5

(Remarks to the Author)
The authors adequately addressed all my concerns. There are no further suggestions.

(Remarks on code availability)

Point-by-point Response to Reviewers' Comments (in blue)

We would like to thank all the three reviewers for their invaluable suggestions and comments that help improve the quality of this manuscript. We have carefully revised our manuscript accordingly. The point-by-point response to the comments are listed below.

Reviewer #1

The authors propose a new pharmacophore fingerprint constrained molecular de novo design method, which they are also experimentally validating.

There are several pharmacophore based molecular de novo generators out there, but this one has innovative aspects that makes the manuscript potentially suitable for publication.

However, the comparison with other methods is insufficient, since the other methods are using 3D pharmacophores and not pharmacophore fingerprints. It would be more relevant to do a comparison with the same pharmacophore fingerprint used here implemented in a platform like REINVENT.”

Response: We appreciate the Reviewer's suggestions. While it is theoretically possible to use REINVENT with pharmacophore similarity (to a specific compound) as the reward function to generate molecules similar to the target compound in terms of pharmacophore, its efficiency is lower than conditional generative models like TransPharmer and other benchmarked methods, since a new REINVENT model have to be trained independently through reinforcement learning for each of the 300 target compounds in the test set that takes a long time. As TransPharmer and REINVENT fall into different categories of molecular generative models, we did not carry out direct comparison in this study.

Furthermore, optimizing model architecture is beyond the major scope of this work. The methodological contributions of our work lie in the use of a simple yet informative topological pharmacophore fingerprint to guide molecule generation. Few studies have explored the potential of topological pharmacophores in generative models. The most relevant work might be RG2Smi, introduced by scientists at Lilly¹, which uses a SMILES-like representation of a reduced graph to guide molecule generation. However, RG2Smi is not open-sourced and has not been tested in real drug design scenarios publicly.

I don't agree that the new PLK1 inhibitors are especially novel. The left side and the hinge binding region is the same as Onvensartib. I would expect larger differences between the new compounds and onvensartib to think this would be an interesting example.

¹ Pogány, Peter, et al. "De novo molecule design by translating from reduced graphs to SMILES." *Journal of chemical information and modeling* 59.3 (2018): 1136-1146.

Response: Onvansertib is characterized by its pyrazolo-quinazoline core, which is patented (e.g., WO2008074788). Therefore, our approach focused on identifying novel and functional core scaffolds while allowing for moieties similar to 1-phenylpiperazine on the left side of Onvansertib during cherry-picking and chemical synthesis. Importantly, the distinct core scaffolds of the new compounds can lead to significant differences in pharmacokinetic properties. Our preliminary in vivo pharmacokinetic experiments in mice demonstrate that IIP0943 has a prolonged half-life and over two-fold oral bioavailability compared to Onvansertib. Below, we present a comparison of some pharmacokinetic properties of IIP0943 with two clinical-stage inhibitors.

Compound	$t_{1/2}$ (h)	C_{max} (ng/mL)	AUC_{∞} (h·ng/mL)	F (%)
IIP0943	3.1	1001	5306	59
Onvansertib ¹	1.6	341	1086	24
Volasertib ²	(Not reported)	(Different dose)	(Different dose)	41

¹Data of Onvansertib from Beria I. et al. *Bioorg. Med. Chem. Lett.* 21.10 (2011): 2969-2974.

²Data of Volasertib from Rudolph D. et al. *Clin. Cancer Res.* 15.9 (2009): 3094-3102.

Thus, the similarity in the left side and the hinge-binding region to Onvansertib is primarily due to the conservative strategies we adopted during manual selection and chemical synthesis. We emphasize that TransPharmer is capable of generating entirely distinct and promising compounds. Below are examples of other novel compounds generated by TransPharmer in the same round as IIP0943 (can also be found in Supplementary Figure 2 and 3) and in another round of our in-house design.

As is clear from EXCAPE-DB, there exists already several thousand PLK1 inhibitors so there is no scientific novelty in publishing more.

Response: Although there are many PLK1 inhibitors recorded in EXCAPE-DB, the majority of

these compounds exhibit low inhibitory activities (only about 15% have $pXC50 > 6$), and many potent PLK1 inhibitors share similar scaffolds. Therefore, we believe it remains scientifically significant to identify new series of potent and selective PLK1 inhibitors with novel scaffolds.

I don't see much point in using a pharmacophore finger print for finding novel PLK1 inhibitors, a docking protocol to score the generated molecules would be more suitable.

Response: We would like to make it clear that TransPharmer is pharmacophore-based generative model. And pharmacophore fingerprints have been effectively used in virtual screening to achieve scaffold hopping of known active ligands². We believe that pharmacophore-based generative models are valuable alternatives to methods like pharmacophore fingerprints- or docking-based virtual screening for discovering novel active compounds. In our work, we have used a docking protocol to score the generated molecules. More importantly, our findings demonstrate that the synergy between these methods can lead to meaningful and efficient discoveries in drug design.

² Such as: Schneider, Gisbert, et al. "Scaffold-Hopping" by Topological Pharmacophore Search: A Contribution to Virtual Screening." *Angewandte Chemie International Edition* 38.19 (1999): 2894-2896;

Stiefl, Nikolaus, et al. "ErG: 2D pharmacophore descriptions for scaffold hopping." *Journal of chemical information and modeling* 46.1 (2006): 208-220.

Reutlinger, Michael, et al. "Chemically advanced template search (CATS) for scaffold-hopping and prospective target prediction for 'orphan' molecules." *Molecular informatics* 32.2 (2013): 133;

Reviewer #2

The authors have designed a novel method to generate molecules conditioned on 2D pharmacophores. To generate the datasets they extract pharmacophore features using functions from RDKit and define the inter-feature distance based on the topological distance between them. The model itself is a sequence based transformer that takes pharmacophore fingerprints as conditional input. The authors show that the model generates molecules that are well conditioned on the pharmacophore input and can be a useful tool for real world discovery. They further provide two case studies on the DRD2 and PLK1 receptor systems to showcase its utility in drug discovery projects. I believe the proposed method is impactful and would be valuable to readers of this domain and hence should be published. I have, however, a few comments that may improve the quality of the publication:

Minor Comments:

The comparison with PGMG in Supplementary Table 1 and 2 is confusing:

The authors on page 6-7 mention that the TransPharmer closely approximates the match scores of PGMG but in Supplementary Table 1 it is very clear that PGMG does much better than TransPharmer, I would not say the numbers are a “close approximation” of each other.

Response: Thank you for your comment. We agree that though the difference in match scores between TransPharmer-108bit and PGMG is less than 10%, the differences of the two other models are about 15% that may be inappropriate to describe as “close approximation”. We have revised the original statement to better highlight their differences in the revised manuscript: “Consequently, we re-evaluated TransPharmer based on the match score utilized in PGMG and found that the match scores achieved by TransPharmer are a little lower than that of PGMG, with about 10% difference for the TransPharmer-108bit model and about 15% for the other two models.”

The authors need to describe the experiments in supplementary 2 in more detail. Why were 114 molecules selected? Why is it important to evaluate molecules with 2 distinct conformers for this experiment? Are the other rows in the table on the entire test set? Furthermore, why are some of the average deviation numbers negative? Should deviation not always be a positive value? Finally, what's the final takeaway message from these results? - The authors should add another paragraph or 2 explaining this more.

Response: Thank you for your comments and suggestions. We noted that PGMG derives its 3D pharmacophore from a molecule embedded in 3D space. In our benchmarking, we followed PGMG's guidelines, using RDKit's ETKDG methods to generate a conformation for each sample in the test set. We speculated that PGMG might be sensitive to the different conformations that a molecule may adopt. Therefore, we selected 114 molecules in the test set that exhibit flexible

conformations. A molecule is considered to have a flexible conformation if, after using RDKit's ETKDG to generate ten independent conformations, at least one conformation has an RMSD over 2 Å compared to the first conformation. The performance of PGMG evaluated on these 114 molecules is labeled as "(114 cases, embed1)" for molecules adopting the first conformation and "(114 cases, embed2)" for molecules adopting the conformation with RMSD > 2 Å. Other rows in the table represent results on the entire test set.

Regarding negative deviation values, we define deviation, for example of molecular weight between the generated compound and the reference compound, as the molecular weight of the generated molecule minus that of the reference compound. This definition helps us identify whether the sizes of the generated molecules tend to be larger or smaller than the reference compounds.

We have made the above clearer in the caption of Supplementary Table 2 as well as in a newly added subsection titled "Further evaluation of and comparison with PGMG" in the Supplementary Results.

The final takeaway message from these results is summarized in the main text: "Meanwhile, it is worth noting that PGMG is sensitive to the maximum number of input pharmacophore features specified by users, which leads to a notable deviation in molecular sizes compared to the desired targets, particularly when reference compounds possess flexible conformations."

I believe the paper's quality will also be improved if they did an unconditional generation benchmark on the MOSES dataset as well, as there are many 2D generation model that benchmark on that dataset and it would provide a nice understanding on the molecule generation capability of the model

Response: Thank you for the suggestion. We have included the results of the unconditional generation benchmark on the MOSES dataset in the Supplementary Results under the subsection "Benchmarking the unconditional TransPharmer and other evaluations". In summary, the unconditional TransPharmer ranks in the top 2 places among other compared models in six out of 15 metrics benchmarked in MOSES.

Is the precision numbers in Supplementary Table 6 based on only 4000 molecules? It would be useful to mention that explicitly in the table caption.

Response: We appreciate the Reviewer for pointing out this oversight. The precision numbers in Supplementary Table 6 are indeed based on 4000 samples. We have now corrected the table to explicitly indicate the number of samples used to obtain the precision numbers.

Page 12 - It would be useful to expand on DeepDrugCoder, is the predictive model same as the QSAR model? And what is the reserved set mentioned here?

Response: We have detailed the settings of the predictive model in the Methods section under the subsection "DRD2 QSAR model." We adhered to DeepDrugCoder's designation of the predictive model as a QSAR model. The "reserved set" refers to the test set unseen by the QSAR model during

training and is used by the FPB model in DeepDrugCoder to guide molecule generation.

Finally, there is a lot of useful content in the supplementary information that is not referenced in the main text, it may help the reader get more context about the method if those experiments are discussed briefly/ referenced in the main text.

Response: We have now added several paragraphs at the beginning of the Results section to include references of contents in the supplementary information. This should give readers a more comprehensive understanding of the model's performance.

The code is not accessible.

Response: We sincerely apologize for the inconvenience regarding access to the code repository. The source code is now accessible at <https://github.com/iipharma/transpharmer-repo>, and the "Code availability" section in the main text has been updated accordingly. The code repository contains detailed guidelines for reproducing the results presented in the manuscript.

Reviewer #3 & #4

In the manuscript, “Accelerating Discovery of Novel and Bioactive Ligands With Pharmacophore-Informed Generative Models” by Xie et al., the authors describe the development of an innovative generative model to identify not only novel but also bioactive compounds. They described a 2D pharmacophore-based fingerprint approach for the generation of structurally diverse but pharmacophorically similar molecular structures. Their aim was to provide suggestions to explore the active chemical space for a target and achieving a high probability of activity for the generated compounds. The approach is innovative and makes use of one of the main advantages of pharmacophore-based similarity searches: scaffold hopping. In general, a very interesting study.

While the benchmarking experiments and retrospective study on DRD2 ligands showed promising data but also already suggest some of the challenges of this approach, the experimental case study somehow confirms some limitations of this method in real life projects. Actually, the case study is the main reason why I think this work is exceptional and very informative: In their case study on PLK1 inhibitors, TransPharmer was used to generate novel structures as suggestions for synthesis. The model returned numerous structures, however, many of them are from the same scaffold as training compounds, includes duplicates, non-drug-like structures etc. Using additional filters, the suggested structures could be narrowed down to the most promising ones.

The virtual screening part included two docking runs and MD simulations. Unfortunately, the validation of this workflows has not been described.

The 42 top-ranked molecules were further inspected and the final selection of compounds for synthesis was based on synthesizability, novelty, and the diversity of the generated compounds. Anyways, the finally synthesized compounds slightly differed from the computed structures. Notably, the 2,6-dimethylpiperazine moiety was replaced by 1-methylpiperazine, which is also found in onvansertib, making it more similar to known active compounds.

The authors further included an enzyme activity as well as a proliferation assay in the human colon cancer cell line HCT116 to verify the computationally generated hits.

In the end, the team succeeded in the development of a highly active, structurally novel compound and two other compounds with moderate activity.

In general, we would recommend rereading the whole manuscript concerning spelling mis-takes, gaps between numbers and the corresponding unit and grammar guides especially in the methods section.

Response: We apologize for the inconvenience. We have thoroughly proofread the manuscript to correct spelling mistakes, missing gaps between values and units, and other grammatical errors.

- describe the validation of the docking and MD runs

Response: We validated our docking protocol and MD runs by redocking Onvansertib into the ATP pocket of PLK1 and running a 100 ns MD simulation using the redocked complex as the starting structure. Onvansertib was favorably scored by Glide-XP with a docking score of -11.58 kcal/mol, and the RMSD between its No.1 docking pose and the co-crystal pose was less than 0.5 Å. The RMSD of Onvansertib in the last 20 ns of the entire MD trajectory, after aligning the protein structures, was 1.71 Å. We have included this validation description of the docking protocol and MD runs in the Methods section.

- please describe how synthesizability was assessed

Response: We assessed synthesizability using the Synthetic Accessibility (SA) score³ to estimate the ease of synthesis of the designed compounds, supplemented by manual inspection by medicinal chemistry experts. These are explicitly mentioned in the “Chemical Synthesis” subsection of the Methods section now.

- please include C13 NMR spectra of the newly synthesized compounds in the supporting information

Response: We have included the C13 NMR spectra of the newly synthesized compounds in the supporting information.

- how did the actually synthesized compounds perform in the virtual screening?

Response: The actually synthesized compounds performed comparably or better in the virtual screening compared to the generated structures, particularly for those with good bioactivities. Below is a detailed comparison of their performance in docking, MM/GBSA estimation, and MD simulations:

³ Ertl, Peter, and Ansgar Schuffenhauer. "Estimation of synthetic accessibility score of drug-like molecules based on molecular complexity and fragment contributions." *Journal of cheminformatics* 1 (2009): 1-11.

Compound	Glide XP score (kcal/mol)	MM/GBSA ΔG (kcal/mol)	RMSD (Å, last 20 ns)	Interaction frequency in 100 ns MD			
				Lys82	Cys133	Glu140	Asp194
lig-3	-12.5366	-58.80	1.60	0.940	1.000	0.891	0.841
IIP0942	-12.1702	-61.76	1.64	1.000	1.000	1.000	0.910
lig-182	-11.9427	-54.24	2.44	0.821	1.000	0.673	0.980
IIP0943	-11.0564	-64.71	1.87	1.000	1.000	0.851	0.821
lig-886	-11.5964	-62.19	1.26	0.465	1.000	0.049	1.000
IIP0944	-11.0212	-54.89	1.74	0.554	1.000	0.534	0.663
lig-524	-11.5255	-60.48	1.55	0.891	1.000	0.990	0.584
IIP0945	-10.8228	-67.13	1.46	0.851	1.000	1.000	0.405

- clearly address points of bias in the case study on PLK1 inhibitors and how they may have influenced the results

Response: We think the following aspects might introduce biases that could affect the results in the case study of PLK1 inhibitors:

a. During molecule generation:

a) Input pharmacophore fingerprint/reference compound. Since TransPharmer is a conditional generative model, the choice of input condition (pharmacophore fingerprint of the reference ligand) could be the largest source of bias in this work. We selected Onvansertib as the reference ligand for its potency and high selectivity towards PLK1, as well as its recent activity in clinical trials. We also used its follow-up derivatives (compound **13** and **25**)⁴ as inputs for TransPharmer to generate compounds in our in-house tests and observed slight variations in chemical space coverage (visualized by t-SNE plots). We surmise that using pharmacophore fingerprints from other unlike PLK1 inhibitors would result in significant differences in generated compounds.

b) Model hyperparameters. One key hyperparameter is the sampling temperature (t). This parameter re-weights the multinomial distribution of each token in generated SMILES strings, with lower temperatures increasing the probability of the top-ranked tokens relatively. In our tests, a higher sampling temperature (t=1.2) improved diversity and novelty but also significantly reduced the performance of top-ranked compounds compared to a default lower temperature (t=0.7). We expect there are some balance points to improve diversity without losing much performance. We stuck to the sampling temperature at 0.7 in our work, but users may explore this hyperparameter further in their own studies by adjusting it in the configuration file.

b. During virtual screening:

a) Novelty filters. Although TransPharmer can generate novel compounds,

⁴ Caruso, Michele, et al. "5-(2-amino-pyrimidin-4-yl)-1H-pyrrole and 2-(2-amino-pyrimidin-4-yl)-1, 5, 6, 7-tetrahydro-pyrrolo [3, 2-c] pyridin-4-one derivatives as new classes of selective and orally available Polo-like kinase 1 inhibitors." *Bioorganic & medicinal chemistry letters* 22.1 (2012): 96-101.

chemical structures that are highly similar to the reference compound also appeared in the generated set as they easily fulfilled the target pharmacophore condition. To avoid overrepresentation of these similar compounds among top-ranked compounds, we used SMARTS patterns to retain structurally novel compounds. In the PLK1 case study, we used the pattern c12ncncc1-CC-[n,c]3:[n,c]:[n,c]:[n,c]:[n,c]32 to filter out molecules with scaffolds similar to the pyrazolo-quinazoline core of Onvansertib. This novelty filter was very effective, but a different SMARTS pattern may significantly impact the results.

- b) Target binding mode. Since we used Onvansertib to provide the input pharmacophore fingerprint, we focused on four polar interactions between Onvansertib and PLK1: hydrogen bonds with Cys133, Lys82, and Asp194, and a salt bridge with Glu140. Generated compounds were scored based on the occurrence of these interactions in their docked complex with PLK1, with a mandatory requirement for forming hydrogen bonds with Cys133. Different scoring criteria might yield different outcomes.
- c. During manual inspection: After virtual screening, a ranked list of 2300 generated compounds were cherry-picked to retain 42 promising compounds for further evaluation. We specifically focused on some aspects of Onvansertib during our visual inspection and cherry-picking. These may constitute potential biases in the case study of PLK1 as well.
 - a) Core region. One objective of the PLK1 case study was to identify compounds with distinct scaffolds from the pyrazolo-quinazoline core moiety of Onvansertib, which is patented (such as WO2008074788). During manual inspection of the generated compounds, we prioritized novel scaffolds while tolerating those containing Onvansertib's 1-phenylpiperazine moiety.
 - b) 3D shape. When inspecting the docking poses, we tended to select compounds adopting a similar U-shape pose to Onvansertib (PDB ID: 2YAC), although we also considered promising molecules with different binding poses (e.g., L-shape).

The above discussions have been included in the Discussion section under the subsection "Potential biases in the case study of PLK1 inhibitors design".

- in all graphs and the text, replace dose-response with concentration-response. Dose refers to e.g. mg/kg and is applicable to in vivo data only.

Response: We appreciate the Reviewer's reminder. We have replaced "dose-response" with "concentration-response" in all graphs and throughout the text.

- we recommend presenting the IC50 values in a separate table in order to ensure clarity. Please explain how the IC50 values were calculated. Which statistics were used for enzyme activity as well as cell proliferation assay? Please add the exact method behind.

Response: We appreciate the Reviewer's thoughtful suggestions. We have revised Figure 5 and made a separate table (Table 2) to present the IC50 values in the selectivity assays.

IC50 values were calculated using Prism 8 by fitting the following equation:

$$Y = \text{Bottom} + (\text{Top} - \text{Bottom}) / (1 + 10^{(\log \text{IC}_{50} - X) * \text{HillSlope}})$$

where X is log of concentration, Y is response, and Top and Bottom are the responses of controls. Each assay was repeated at least three times, and we computed the mean and standard deviation for the values. The description of IC50 calculation has now been included in Methods under the subsection “In vitro kinase activity assays”.

- Please show the confidence interval (CI) of the IC50 values instead of the standard deviation or as additional information, since the CI in this case is much more significant.

Response: We have added the computed confidence intervals (CIs) of the IC50 values for enzyme activity against PLK1 in Supplementary Table 7, which is also shown below.

Compound	90% CI (nM)	95% CI (nM)	99% CI (nM)
IIP0942	37.57±3.55	37.57±4.23	37.57±5.56
IIP0943	5.06±1.64	5.06±1.96	5.06±2.57
IIP0944	—	—	—
IIP0945	927.7±122.5	927.7±145.9	927.7±191.7
Onvansertib	4.80±0.65	4.80±0.77	4.80±1.01

- Some measurement points in Figure 5b, Figure 6a and Supplementary Figure 4 have error bars, whereas others do not. Especially for cell-based assays, this is impressive.

Response: We have reviewed the experimental data and found that the measurement points “without error bars” actually have relatively much smaller error bars, making them appear invisible in the figures. We have attached the raw data file of the cell-based assays (“raw_data_HCT116_iip0943_onv.pdf”) for the reviewers to examine.

- It would also be interesting to include the IIP0942 compound in the cell proliferation assay conducted in HCT116 cells. Although the substance seems not to be as effective in inhibiting PLK1 enzyme activity as IIP0943, it would be great to see the impact of the compound in a physiological context. One may also add volasertib as another PLK1 inhibitor with much higher protein affinity than onvansertib for comparison reasons.

Response: We have tested the inhibitory activities of IIP0942 and Volasertib in the cell proliferation assay of HCT116 cells. The results are shown in the figure below and added as the Supplementary Figure 4. We selected Onvansertib as the main reference compound instead of Volasertib because, in our in-house experiments, these two inhibitors are comparable in both enzymatic (0.67 nM vs. 0.30 nM) and cellular activity (HCT116, 36 nM vs. 42 nM). However, Volasertib also exhibits low nanomolar activities against PLK2 and PLK3, resulting in much worse selectivity compared to Onvansertib.

Concentration response on HCT116

- The protocol for in vitro kinase activity assays and cell viability assays is traceable. However, were the cells really seeded into U-bottom plates?

Response: We have consulted our assay service provider and were informed that they used Corning® 96-well plates, which are designed to prevent cell adhesion on U-bottom plates.

- It would also be useful to add the effect of the DMSO control compared to the compounds in the supplement. As stated above, please add the underlying statistical analysis in the methods section for the IC_{50} value calculation.

Response: While we did not include the effect of the DMSO control in the figures to allow for direct comparison with other compounds, we are pleased to provide the raw data file (“raw_data_HCT116_iip0943_onv.pdf”) for reviewers’ examination. We will also include this information in the supplementary materials for formal publication if required. The issue concerning the IC_{50} value calculation has been addressed above in response to question #8.

Point-by-point Response to Reviewers' Comments (in blue)

Reviewer #1 (Remarks to the Author):

The authors haven't addressed any of my objections in a satisfactory way, I struggle to see the importance of the work to justify publication in Nature Communications.

Response:

We would like to take this opportunity to provide additional clarification on the reviewer's previous comments on the comparison with REINVENT: "However, the comparison with other methods is insufficient, since the other methods are using 3D pharmacophores and not pharmacophore fingerprints. It would be more relevant to do a comparison with the same pharmacophore fingerprint used here implemented in a platform like REINVENT."

We implemented pharmacophore-based REINVENT models (named "REINVENT (PharmFP)") using the pharmacophore fingerprint Tanimoto similarity as the reward during the reinforcement learning stage. More specifically, we fine-tuned a REINVENT prior model, which had been pretrained on the ChEMBLv32 database for 20 epochs, to generate molecules with high similarity to the target compound based on the 72-bit pharmacophore fingerprint as defined in the manuscript.

For evaluation, we applied the REINVENT (PharmFP) to the same test set used to benchmark other conditional generative models, consisting of 300 molecules providing target pharmacophores. The REINVENT (PharmFP) for each molecule in the test set were trained for 1,000 steps during reinforcement learning. Afterward, the agents at the final step were used to sample 1,000 molecules, which were then compared to those generated by TransPharmer-72bit (TransPharmer conditioned on the same 72-bit pharmacophore fingerprint). As summarized in Table R1, TransPharmer outperforms REINVENT (PharmFP) by generating more compounds with higher pharmacophoric similarity and lower deviation in the number of pharmacophore features.

Table R1: Performance on generating compounds with similar pharmacophore.

Model	S_{pharma}	D_{count}
REINVENT (PharmFP)	0.41 ± 0.12	6.1 ± 2.2
TransPharmer-72bit	0.50 ± 0.14	4.6 ± 2.6

S_{pharma} follows the same definition as in the manuscript, calculating the Tanimoto coefficient of the ErG fingerprints. D_{count} computes the average deviation of the amount of pharmacophore features, as defined in the manuscript.

We also applied REINVENT (PharmFP) to the case study of designing PLK1 inhibitors, using Onvansertib as the target compound, as demonstration. The reinforcement learning curve is shown in the Figure R1 below.

Figure R1: The learning curve of REINVENT (PharmFP). The reward (y axis) reaches plateau at the end of the reinforcement learning.

Table R2: Performance on generating compounds with similar pharmacophore to Onvansertib.

Model	S_{pharma}	$S_{\text{pharma}} (72\text{bit})$	D_{count}
REINVENT (PharmFP)	0.491	0.805	5.848
TransPharmer-72bit	0.621	0.914	4.757

$S_{\text{pharma}} (72\text{bit})$ computes the Tanimoto coefficient of the 72-bit pharmacophore fingerprints, which is also the reward function of REINVENT (PharmFP).

REINVENT (PharmFP) required approximately 53 minutes (1,000 steps) to converge for this single goal, whereas TransPharmer, once trained, was able to generate the same number of molecules within 1 minute. As shown in the Table R2, the molecules generated by TransPharmer aligned with the target topological pharmacophore better, exhibiting higher S_{pharma} and $S_{\text{pharma}} (72\text{bit})$ scores, and a lower D_{count} . The low efficiency of REINVENT (PharmFP) was expected, as reinforcement learning can take long time to explore the chemical space and the model may sometimes fall into suboptimal solutions. In contrast, conditional generative models like TransPharmer demonstrate greater efficiency in sampling molecules that satisfy the target goals.

These additional results have been included in the Supplementary Section “Comparison with REINVENT”.

As for the reviewer’s other comments, we thought that we have fully addressed them in the previous revised manuscript and explained in the response letter. We could provide further clarification if needed.

Reviewer #2 (Remarks to the Author):

No more comments

Reviewer #3 (Remarks to the Author):

The authors have addressed the comments accordingly. In its current form, it can be published.

Reviewer #4 (Remarks to the Author):

Reviewer #4 (Remarks on code availability):

The code includes all necessary information about installing and running the TransPharmer application.

Reviewer #5 (Remarks to the Author):

In this study, the authors proposed TransPharmer, a de novo molecule generation model that integrates pharmacophore fingerprints with the generative pre-training transformer. From a purely methodological/technical perspective, novelty is limited; however, the execution and the molecular analyses are quite good. I also evaluated the reviewer reports for the first round and the authors' responses. Below, I list my specific concerns.

1)

The manuscript lacks molecular novelty-based analyses. It is important to generate de novo molecules with similar pharmacophore-centric features to known effective inhibitors of the selected target for functionality; however, another important feature of generative AI models is the novelty (not only against the guiding molecules but also against the entire known chemical space). This is important to assess the model's (or we can say the proposed approach's) usefulness when utilised in real-life drug design/development pipelines. I'm not purely talking about the MOSES/GuacaMol benchmarking metric called novelty; instead, I'm talking about assessing the novel features of molecules in general (both qualitatively and quantitatively).

Response: We greatly appreciate the reviewer's insightful comments. We fully acknowledge the

importance of novelty in *de novo* molecule generation. While conducting a comprehensive novelty assessment against the entire known chemical space can be a time-intensive process, it is more conventional and practical to focus on compounds with predicted or validated bioactivity. As suggested by the reviewer, we computed the nearest neighbor similarity to all known DRD2 active ligands in the ExCAPE-DB database for each generated compound predicted to be active against DRD2 by our QSAR model described in the manuscript. The distribution of the nearest neighbor similarities is shown in Figure R2, where approximately 43% of the compounds exhibit a similarity score of less than 0.4.

Figure R2: The distribution of nearest neighbor similarity of the generated compounds to known DRD2 actives.

In our previous manuscript, we have carried out a novelty assessment for the identified hits in the PLK1 case study, specifically at three levels: (1) within known PLK1 actives, (2) within reported bioactive ligands, and (3) within patented compounds. Our findings indicate that the identified hit compounds, especially IIP0943, represent novel chemotypes of selective PLK1 inhibitors, which underscores the practical value of our approach in real-world drug design. These results are presented in the Supplementary Section “Molecular novelty assessment of the discovered hits,” and we have added a reference to this section at the end of the “Case Study of PLK1” for further clarity.

2)

There is no ablation study. It is important to assess which design choices led to the obtained results. One obvious comparison would be between TransPharmer and a model with the exact same ML architecture that does not utilise the pharmacophore fingerprints.

Response: We thank the reviewer for the valuable suggestion. In response, we have conducted an ablation study to evaluate the impact of different aspects in the pharmacophore fingerprint in guiding the exploration of chemical subspaces of interest. Specifically, we compared our 72-bit model with two TransPharmer variants that omit key features from the pharmacophore fingerprint: topological distance information and feature combination. Additionally, we compared an unconditional variant of TransPharmer, which does not utilize the pharmacophore fingerprints or any pharmacophore information. The results of this ablation comparison are summarized in the following Table R3. In brief, TransPharmer demonstrates a more efficient navigation toward the relevant chemical space when guided by the pharmacophore fingerprints, compared to the unconditional variant. And the

topological distance information and feature combination in the pharmacophore fingerprint significantly enhance TransPharmer’s performance.

The details of the ablation study are now included in the Supplementary Section titled “Ablation study” and the results are listed in the Supplementary Table 15 (and the Table R3 below).

Table R3: Results of ablation study.

Model	De novo generation		Scaffold elaboration	
	D _{count}	S _{pharma}	D _{count}	S _{pharma}
TransPharmer	4.6±2.6	0.50±0.14	3.0±2.2	0.702±0.176
without topo*	6.3±2.6	0.38±0.10	3.9±2.2	0.609±0.206
without topo and combo**	8.5±2.7	0.31±0.08	5.3±2.7	0.547±0.223
without any guidance***	10.5±2.5	0.27±0.07	6.6±3.6	0.514±0.239

Bold numbers indicate the best results. S_{pharma} and D_{count} follow the same definitions as in the main text.

* omitting topological distance information;

** omitting topological distance information and feature combination;

*** no pharmacophore guidance (unconditional generation).

3)

In today’s AI research, the interpretability of generative models is an important topic. The authors may consider an explainability-centric analysis to observe and understand what the model has learned in the context of molecular properties (e.g., using attention maps). The current state of the section entitled "Controllability of TransPharmer" does not qualify for this, but additional evaluations and discussions can solve this.

Response: We agree that understanding the mechanisms of deep learning models is both important and challenging. The section titled “Controllability of TransPharmer” is an initial attempt to address this by exploring how well each bit of the pharmacophore fingerprint controls molecule generation. As suggested by the reviewer, we examined the attention maps across all transformer blocks and attention heads, though, unfortunately, the results were not informative. We were unable to identify clear qualitative relationships between individual pharmacophore fingerprint bits and specific molecular moieties. Most attention maps appeared sparse, and while we did observe some densely activated maps in a few attention heads within the middle layers (shown in the Figure R3 below), these observations were limited and the patterns were obscure.

The interpretability of deep learning models, especially of generative models, is an unsolved challenge and requires extensive exploration. We believe that enhancing the interpretability of generative models, and aligning them more closely with human intuition, may require incorporating stronger inductive biases into the model architecture. We view this as an important direction for future work.

Figure R3: The attention maps of an exemplar molecule shown in the left. Most attention maps are sparse, except for a few attention heads in Layer 2, such as Head 7.

4)

It looks like there is something wrong with the t-SNE output in Figure 4a; 10,000 random training molecules and 3,873 PLK1 active ligands occupy nearly the exact same molecular space, whereas PLK1 active ligands should have occupied a specific sub-region of the random molecule space. As a result, comments made about the space occupied by the generated compounds are not valid as well. This is probably related to the hyperparameters selected for the t-SNE projection. This analysis should be repeated with different parameters to obtain a more meaningful result, and the corresponding discussion should be updated accordingly.

Response: We sincerely thank the reviewer for pointing out this issue. We have replotted the t-SNE projection using pharmacophore fingerprints instead of common physicochemical descriptors to produce a more meaningful result, as shown in the revised Figure 4a (left figure in the Figure R4 below). Interestingly, while the PLK1 active ligands still overlap with the chemical space of the training set, the generated compounds occupy a distinct subspace. We speculate that this overlap is due to the fact that the training set contains many kinase inhibitors, and most PLK1 active ligands are not highly selective. This may explain the relatively small differences in the chemical spaces of the training set and the PLK1 active compounds.

We also found that Onvansertib behaves as an “outlier” relative to most PLK1 active ligands and the training set (as shown in the right figure in the Figure R4 below). The unconditional TransPharmer generates molecules that cover a similar chemical space to the training set. However, when conditioned on the topological pharmacophore of Onvansertib, the chemical space of the generated compounds shifts effectively toward regions surrounding Onvansertib. This observation illustrates TransPharmer’s ability to explore the boundaries of the learned chemical space and further addresses the reviewer’s first concern regarding the novelty of the generated compounds from the view of chemical space projection.

We have updated the corresponding discussion and methodology in the revised manuscript to reflect these findings.

Figure R4: (left) The t-SNE plot of molecules from the training set, the unconditionally generated set (Generated set (uc)), the conditionally generated set and known PLK1 actives. (right) The t-SNE plot without generated sets. The position of Onvansertib is highlighted by the red arrow.

Point-by-point Response to Reviewers' Comments (in blue)

We would like to thank all the reviewers for their invaluable suggestions and comments that help improve the quality of this manuscript. We have carefully revised our manuscript accordingly. The point-by-point response to the comments are listed below.

Reviewer #1 (Remarks to the Author):

The authors has done enough to convince me that the study is publishable.

Reviewer #5 (Remarks to the Author):

I thank the authors for their efforts. However, my concerns were only partially addressed with this revision. I'm providing detailed information regarding the missing bits below (I'm using the same numbering for the issues in my original review, considering reviewer #5).

1)

I could not find the newly added text and figure about the novelty analysis regarding the nearest neighbour similarity of the generated compounds to known DRD2 actives, neither in the main text nor the supplementary material of the revised manuscript. Where is it? It should certainly be included in the manuscript. Moreover, the information regarding this new novelty analysis in the response to the reviewers' document contains no discussion or insight. It only states, "43% of the compounds exhibit a similarity score of less than 0.4". This is insufficient. Please make the necessary modifications and additions.

Response: Thank you for the suggestion. We have now included the novelty analysis of the generated compounds against DRD2 and added more discussion at the end of the Section "Case study of DRD2" (highlighted in red in the revised manuscript) as: "... For molecules predicted to be active but not previously identified as DRD2 actives, we assessed structural similarities to their nearest DRD2 active neighbor. The similarity score distribution peaks around 0.4, with 43% of the molecules having similarity score below 0.4, a commonly used threshold for classifying dissimilar compounds (Supplementary Figure 2). This suggests a high degree of structural novelty among the generated molecules compared to known DRD2 actives. Overall, these findings highlight TransPharmer's capability to both rediscover known active ligands and to generate structurally novel compounds with potential bioactivity."

2)

An ablation study has been added to the supplementary material but has not been mentioned in the main text. This mention, together with a summary of the results obtained from the ablation study, should be added to a relevant section in the main text.

Response: We are sorry for not including the related information in the main text. We now include description about the ablation study results in the Section “Pharmacophore-constrained molecule generation” in the revised main text (highlighted in red) as: “... An ablation study further showed that the incorporation of topological distance information and feature combinations into the pharmacophore fingerprint substantially contributes to TransPharmer's overall performance (Supplementary Results “Ablation study” and Supplementary Table 15). For the *de novo* generation task, removing the topological distance information decreased the pharmacophore similarity score from 0.50 to 0.38 and removing both the topological distance information and feature combinations further decreased it to 0.31. For the scaffold elaboration task, removing both the topological distance information and feature combinations decreased the pharmacophore similarity score from 0.70 to 0.55.”

3)

Similar to issue number 1, the analyses conducted to address this issue have not been included anywhere, not in the main text or the supplementary material. Even though the results of the analysis were inconclusive, the findings should be reported and discussed in the manuscript. It is important to discuss the possible reasons behind not being able to identify clear qualitative relationships between individual pharmacophore fingerprint bits and specific molecular moieties. By referring to the figure, it would be good to mention and discuss those densely activated maps in a few attention heads within the middle layers.

Finally, a future work paragraph should be added to the conclusion section, in which information regarding “more detailed work is to be conducted in this scope” should be mentioned. Other possible future work should also be discussed in that paragraph as well.

Response: Thank you for the constructive suggestions. We now include the analysis and additional discussion of attention maps in the revised manuscript under the Section “Model controllability and interpretability” (replacing the original “Controllability of TransPharmer”; highlighted in red) and add a new supplementary figure that shows the attention maps of an exemplar compound (Supplementary Figure 6, which is also included below):

“To further investigate what the models have learned, we analyzed the attention maps across all transformer blocks and attention heads. Though most attention maps appeared sparse, particularly in higher layers (Supplementary Figure 6a), we did observe meaningful patterns in some densely activated maps. As exemplified in the Supplementary Figure 6b, where the oxygen atom of the ligand, which is the only hydrogen bond acceptor, activates all the corresponding acceptor-related bits in the pharmacophore fingerprint. However, these observations were limited and did not capture all the relationships between each fingerprint bit and the corresponding molecular features as one might expect.

These may be caused by the following reasons: 1) The attention maps may not capture all parts of a generated compound. The linker atoms between pharmacophore features are not expected to activate attention, which could explain the sparsity of the attention maps. The knowledge required to generate proper linkers likely resides in the auto-regressive probabilistic distribution learned by the decoder, but this information cannot be revealed by attention maps. 2) The information from different positions becomes increasingly mixed in higher layers. This accounts for the distinctions observed in the attention maps in the first couple of layers, while in the higher layers, the attention weights tend to be more uniform. These observations are consistent with findings from previous studies and could potentially be addressed by techniques such as attention rollout or attention flow[77]. However, currently there is no universal methods for analyzing attention maps. Even in the well-defined natural language processing tasks, a technique that works well for one task may fail for another[77]. Therefore, fully understanding how generative models learn requires further study.”

We also included discussions of future work to expand the section of “Towards more universal generative models” (replacing the original “Ways to more universal generative models”; highlighted in red): “... Nonetheless, several directions can be explored in the future to enhance the model’s versatility and general applicability. First, additional generation modes, such as fragment-linking, should be incorporated alongside *de novo* generation and scaffold elaboration. Advances in unordered chemical language modeling can directly support these functionalities[80]. Second, generative models that produce easily synthesizable molecules are preferable, as they can accelerate the timeline for wet lab experimental validation[81]. Finally, multi-objective optimization should be integrated into the generative process to support more efficient design. Recent advances, such as integrating Pareto optimization with generative models, may help identify novel compounds with a balanced profile[82].”

Supplementary Figure 6: The attention maps of an exemplar compound. (a) Attention maps across all transformer block and attention heads. (b) The attention map at the 7th head in the second layer.

4)

This one has been addressed.